# FLOW MATCHING CALIBRATION FOR SIMULATION-BASED INFERENCE UNDER MODEL MISSPECIFICATION

## ABSTRACT

Simulation-based inference (SBI) is transforming experimental sciences by enabling parameter estimation in complex non-linear models from simulated data. A persistent challenge, however, is model misspecification: simulators are only approximations of reality, and mismatches between simulated and real data can yield biased or overconfident posteriors. We address this issue by introducing Flow Matching Corrected Posterior Estimation (`FMCPE`), a framework that leverages the flow matching paradigm to refine simulation-trained posterior estimators using a small set of calibration samples. Our approach proceeds in two stages: first, a posterior approximator is trained on abundant simulated data; second, flow matching transports its predictions toward the true posterior supported by real observations, without requiring explicit knowledge of the misspecification. This design enables `FMCPE` to combine the scalability of SBI with robustness to distributional shift. Across synthetic benchmarks and real-world datasets, we show that our proposal consistently mitigates the effects of misspecification, delivering improved inference accuracy and uncertainty calibration compared to standard SBI baselines, while remaining computationally efficient.

## 1 INTRODUCTION

Many fields of science and engineering describe complex phenomena with stochastic models, which capture inherent sources of randomness, such as measurement noise, probabilistic dynamics, etc. While such simulators provide a convenient way to produce synthetic data $\boldsymbol{x} \in \mathbb{R}^d$ given parameters $\boldsymbol{\theta} \in \mathbb{R}^p$, they rarely yield tractable likelihoods, making classical statistical inference methods such as Markov chain Monte Carlo (MCMC) (Robert & Casella, 2005) or variational inference (Rezende & Mohamed, 2015) inapplicable. Simulation-Based Inference (SBI) (Deistler et al., 2025; Cranmer et al., 2020) addresses this limitation by performing Bayesian parameter inference directly from simulated datasets, bypassing the need for an explicit likelihood $p_{\boldsymbol{X}|\boldsymbol{\Theta}}$.

Given a prior $p_{\boldsymbol{\Theta}}$ over parameters and assuming the existence of an unknown real data-generating process $p_{\boldsymbol{Y}|\boldsymbol{\Theta}}$ for observations $\boldsymbol{y} \in \mathbb{R}^d$, SBI algorithms provide various approaches (Papamakarios & Murray, 2016; Lueckmann et al., 2017; Boelts et al., 2022; Hermans et al., 2020) to obtain approximate samples from the posterior distribution $p_{\boldsymbol{\Theta}|\boldsymbol{Y}}$ with the aid of deep generative models. In this paper, we focus on neural-based approaches that directly approximate the posterior distribution, often called neural posterior estimation (NPE). This corresponds to the following form of posterior estimates, where the simulator's likelihood $p_{\boldsymbol{X}|\boldsymbol{\Theta}}$ is used in place of the real data likelihood $p_{\boldsymbol{Y}|\boldsymbol{\Theta}}$,

$$\hat{p}_{\boldsymbol{\Theta}|\boldsymbol{X}}(\boldsymbol{\theta}|\boldsymbol{y}) \propto p_{\boldsymbol{\Theta}}(\boldsymbol{\theta}) p_{\boldsymbol{X}|\boldsymbol{\Theta}}(\boldsymbol{y}|\boldsymbol{\theta}) \ . \tag{1}$$

However, discrepancies between $p_{\boldsymbol{Y}|\boldsymbol{\Theta}}$ and $p_{\boldsymbol{X}|\boldsymbol{\Theta}}$, can severely degrade parameter inference using the above $\hat{p}_{\boldsymbol{\Theta}|\boldsymbol{X}}$. In the Bayesian literature, this problem is referred to as *model misspecification* (Walker, 2013). Misspecification reflects limitations of the simulator or surrogate forward model—whether due to mathematical approximations, simplified dynamics, unmodeled noise, or insufficient computational resources to run more realistic simulations. Such mismatches can lead to biased or overconfident posterior distributions (Frazier et al., 2020; Schmitt et al., 2024), ultimately undermining inference reliability. In particular, modern SBI methods built on deep generative models are especially vulnerable to misspecification. They often perform poorly when faced with out-of-

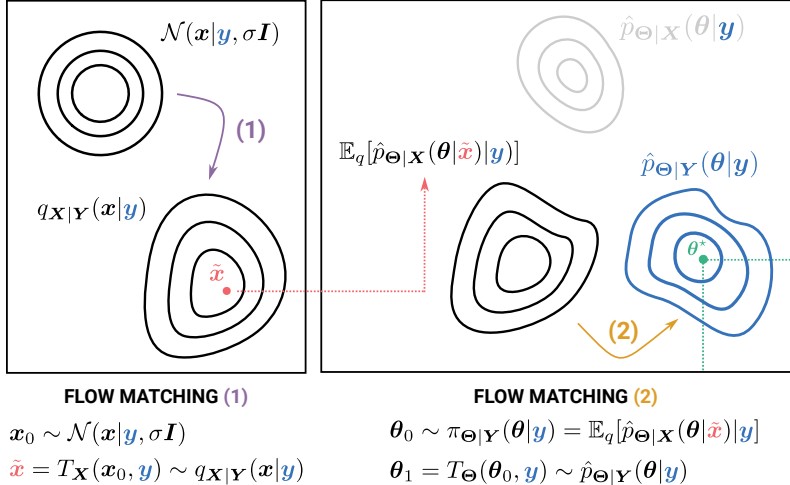

$$x_0 \sim \mathcal{N}(x|y, \sigma I)$$
$$\tilde{x} = T_X(x_0, y) \sim q_{X|Y}(x|y)$$

$$\theta_0 \sim \pi_{\Theta|Y}(\theta|y) = \mathbb{E}_q[\hat{p}_{\Theta|X}(\theta|\tilde{x})|y]$$
$$\theta_1 = T_\Theta(\theta_0, y) \sim \hat{p}_{\Theta|Y}(\theta|y)$$

Figure 1: Overview of FMCPE. The method combines two complementary flow matching steps to correct simulation-based posterior distributions under model misspecification (represented by $\hat{p}_{\Theta|X}(\theta|y)$ in grey level sets). (1) Scarce calibration data $(\theta, y)$ are used to learn a transport map $T_X$ that couples real observations $y$ with surrogate counterparts $\tilde{x}$ lying in the simulator's domain. (2) We then learn $T_\Theta$ to transport samples from a $q_{X|Y}$-weighted version of the simulation-based posterior $\hat{p}_{\Theta|X}(\theta|\tilde{x})$ toward the final corrected posterior $\hat{p}_{\Theta|Y}$. Note that both transports are required: $T_X$ addresses the mismatches between simulated data and real observations, while $T_\Theta$ refines parameter inference to align with the true posterior.

distribution data (Nalisnick et al., 2019) and can fail dramatically across a wide range of real-world problems (Cannon et al., 2022).

A natural way to address imperfections in $\hat{p}_{\Theta|X}$ is to act to reduce its deviation from an estimator built on high-quality data—accurate representations of the phenomenon obtained either from costly high-fidelity simulations or from ground-truth observations. Since such data are typically scarce due to their prohibitive cost, the central challenge is to design strategies that maximally exploit the limited information they provide in order to correct $\hat{p}_{\Theta|X}$.

In this work, we propose a calibration method that leverages the flow matching paradigm of Lipman et al. (2023) to refine posterior estimators trained on simulations using only a small set of costly calibration samples linking parameters with real observations. The principled ability of flows to efficiently model paths between distributions makes them good candidates to design corrective procedures agnostic to the type of misspecifications. Moreover, flows have demonstrated high-scalability with state-of-the-art performance at large-scale image generation (Esser et al., 2024) and data-efficient performance in SBI (Wildberger et al., 2023). Our approach proceeds in two stages. First, a posterior approximator $\hat{p}_{\Theta|X}$ is trained on abundant simulated data via NPE. Second, a reweighted version of this estimator is used to define a proposal distribution $\pi_{\Theta|Y}$, which serves as the source in a flow matching model that learns a transport map to the well-specified posterior $p_{\Theta|Y}$. Crucially, this transport does not require explicit knowledge of the misspecification form: it aligns the simulation-based posterior with the true posterior supported by real observations. The resulting procedure enables accurate estimation of $p_{\Theta|Y}$ despite the limited availability of calibration data. Figure 1 gives an overview of our method, which we call "Flow Matching for Corrected Posterior Estimation" (FMCPE). Experiments on synthetic and real-world tasks show that FMCPE is more robust to misspecification than standard SBI baselines, while being computationally efficient, amortized and applicable as a post-processing to any SBI model.

The remainder of the paper is organized as follows. We begin with an overview of SBI under model misspecification and outline how it relates to our contribution. Next, we motivate and detail our methodology, introducing flow matching concepts and notation as needed. Finally, we present numerical experiments and conclude with a discussion of the results.

## 2 RELATED WORK

Model misspecification in SBI has been studied both in the framework of Approximate Bayesian Computation (ABC) (Frazier et al., 2020; Bharti et al., 2022; Fujisawa et al., 2021) and with modern neural-based approaches (Kelly et al., 2024; Ward et al., 2022; Huang et al., 2023; Schmitt et al., 2024). These works, however, focus on settings distinct from ours. For instance, Ward et al. (2022) assume a known form of misspecification, while Huang et al. (2023) penalize the NPE loss according to the distributional shift between summary statistics of $x$ and $y$. Other contributions, such as Schmitt et al. (2024), restrict attention to detecting misspecification at inference time by imposing a Gaussian prior on the space of summary statistics.

An alternative perspective is to interpret misspecification as a noisy channel between simulated and real-world observations. In this view (Wehenkel et al., 2025; Ward et al., 2022), one assumes that once simulated data $x$ are known, the real observation $y$ does not carry additional information about the parameters $\boldsymbol{\theta}$. Formally, this conditional independence reads

$$y \perp \boldsymbol{\theta} \mid x . \tag{2}$$

Under this hypothesis, the true posterior distribution can be expressed as

$$p_{\boldsymbol{\Theta}|\boldsymbol{Y}}(\boldsymbol{\theta}|y) = \int p_{\boldsymbol{\Theta}|\boldsymbol{X}}(\boldsymbol{\theta}|x) p_{\boldsymbol{X}|\boldsymbol{Y}}(x|y) \, \mathrm{d}x , \tag{3}$$

which states that the true posterior distribution is a mixture of simulation-based posteriors, weighted by the probability of each simulated outcome $x$ given $y$.

Wehenkel et al. (2025) exploit this idea by assuming access to a calibration dataset of parameter–observation pairs $\{(\boldsymbol{\theta}_i, y_i)\}_{1 \le i \le N_{\mathrm{cal}}}$ collected from costly or time-consuming experiments. They approximate $p_{\boldsymbol{\Theta}|\boldsymbol{Y}}$ by replacing $p_{\boldsymbol{X}|\boldsymbol{Y}}$ in Equation (3) with a coupling $q_{\boldsymbol{X}|\boldsymbol{Y}}$ estimated via optimal transport (Peyré & Cuturi, 2019), which links real observations $y$ and simulated data $x$. However, a key limitation is that their method requires access to the full test set at inference time, preventing its use in online or sequential prediction scenarios. Moreover, the conditional independence assumption (2), while convenient, does not always hold. For example, if the model misspecification stems from the fact that the simulator neglects some parameters—say, for $\boldsymbol{\theta} = (\theta_1, \theta_2)$, $p_{\boldsymbol{X}|\boldsymbol{\Theta}}(x|\boldsymbol{\theta}) = f(\theta_1)$ while $p_{\boldsymbol{Y}|\boldsymbol{\Theta}}(y|\boldsymbol{\theta}) = g(\theta_1, \theta_2)$—then $y$ will still depend on the full parameter vector, violating conditional independence.

In contrast, the method MFNPE proposed by Krouglova et al. (2025) does not rely on conditional independence nor require access to the full test set. Their approach first trains a posterior approximator $p_{\mathrm{LF}}(\boldsymbol{\theta}|x)$ on low-fidelity simulations using standard NPE, then refines it with high-fidelity samples (i.e. coming from high-fidelity simulators or real data). Our method follows the same spirit of leveraging low-fidelity approximations obtained from a simulator, but uses them differently: the low-fidelity estimator plays a distinct role, and the subsequent corrections it undergoes are of a different nature, as detailed in the next section.

## 3 FLOW MATCHING CORRECTED POSTERIOR ESTIMATION

We consider the general problem of sampling from a posterior distribution $p_{\boldsymbol{\Theta}|\boldsymbol{Y}}$ resulting from an *unknown* likelihood $p_{\boldsymbol{Y}|\boldsymbol{\Theta}}$ over observation $y \in \mathbb{R}^d$, and known prior $p_{\boldsymbol{\Theta}}$ over a parameter vector $\boldsymbol{\theta} \in \mathbb{R}^p$. If the likelihood were known, the posterior would be given by the Bayes rule as $p_{\boldsymbol{\Theta}|\boldsymbol{Y}}(\boldsymbol{\theta}|y) \propto p_{\boldsymbol{Y}|\boldsymbol{\Theta}}(y|\boldsymbol{\theta}) p_{\boldsymbol{\Theta}}(\boldsymbol{\theta})$. Instead, we assume access to an *imperfect* stochastic simulator $S : (\boldsymbol{\theta}, \epsilon) \mapsto x$, generating low cost simulations $x \in \mathbb{R}^d$, with $\epsilon$ a random noise accounting for randomness in the generative process. The simulator implicitly defines a generative model $p_{\boldsymbol{X}|\boldsymbol{\Theta}}$ whose posterior distribution can be approximated by an easy-to-sample model $\hat{p}_{\boldsymbol{\Theta}|\boldsymbol{X}}$, for instance, using techniques from the SBI literature such as NPE. Unfortunately, since the simulator is inaccurate, the SBI model $\hat{p}_{\boldsymbol{\Theta}|\boldsymbol{X}}$ is likely not to provide an accurate approximation to the true posterior $p_{\boldsymbol{\Theta}|\boldsymbol{Y}}$, in general, no matter how accurately it approximates the simulator's posterior $p_{\boldsymbol{\Theta}|\boldsymbol{X}}$. However, one can reasonably expect such learned model to be informative about the true posterior. We propose to leverage the simulator $S$ and posterior model $\hat{p}_{\boldsymbol{\Theta}|\boldsymbol{X}}$ in addition to a small set of calibration pairs of high-quality data $\mathcal{D}_{\mathrm{cal}} = \{(\boldsymbol{\theta}_j, y_j)\}_{1 \le j \le N_{\mathrm{cal}}}$ from the joint distribution $p_{\boldsymbol{\Theta}, \boldsymbol{Y}}$ to provide an accurate and efficient model $\hat{p}_{\boldsymbol{\Theta}|\boldsymbol{Y}}$ for the true posterior $p_{\boldsymbol{\Theta}|\boldsymbol{Y}}$. Such a dataset typically represents scarce

ground-truth measurements or high-fidelity simulations that are costly to obtain. Consequently, we assume that $N_{\text{cal}}$ is not large enough to provide an accurate posterior estimate from this dataset alone.

We propose to use the flow matching paradigm to learn a dynamic transport map, a vector field, from a carefully designed source distribution $\pi_{\Theta|Y}$ towards the target posterior $p_{\Theta|Y}$ using the calibration dataset $\mathcal{D}_{\text{cal}}$. Increasing the proximity of the source and target distributions reduces the complexity of the flow and makes it easier to learn from a small number of samples. Intuitively, smaller distributional gaps are likely to require fewer steps and improve sample complexity (Cui et al., 2024; Lin et al., 2025; Kong et al., 2025; Wang et al., 2025). Following this principle, we employ the simulator $S$ and the posterior approximation $\hat{p}_{\Theta|X}$, to design and train an informative source distribution that facilitates the learning of the vector field from few calibration samples. The resulting algorithm (Algorithm 1 in Section 3.3) combines simultaneous learning of both the source distribution and the vector field, ensuring the source is continually updated as the vector field becomes more accurate. Next, we detail the approaches for learning the vector field given an informed source distribution, constructing the source itself and training them jointly.

### 3.1 DATA-EFFICIENT POSTERIOR FLOW MATCHING

The goal is to use the calibration dataset $\mathcal{D}_{\text{cal}}$ to learn a transport map $T_{\Theta} : \mathbb{R}^p \times \mathbb{R}^d \to \mathbb{R}^p$ from an easy-to-sample base (or source) distribution $\pi_{\Theta}$ towards the posterior distribution $p_{\Theta|Y}$:

$$\boldsymbol{\theta}_0 \sim \pi_{\Theta}(\boldsymbol{\theta}) \quad \Rightarrow \quad \boldsymbol{\theta}_1 = T_{\Theta}(\boldsymbol{\theta}_0, \boldsymbol{y}) \sim p_{\Theta|Y}(\boldsymbol{\theta}|\boldsymbol{y}) \,. \tag{4}$$

To learn $T_{\Theta}$, we use the flow matching framework of Lipman et al. (2023), an approach that has proven effective for modeling complex target distributions, such as images (Esser et al., 2024), and has recently been applied successfully in simulation-based inference (Wildberger et al., 2023). Flow matching consists of learning a time-dependent vector field $u_{\Theta}$ capable of transporting a sample $\boldsymbol{\theta}$ from $\pi_{\Theta}$ along a trajectory $(\boldsymbol{\theta}_t)_{t \in [0,1]}$ starting from $\boldsymbol{\theta}_0 = \boldsymbol{\theta}$, so that $\boldsymbol{\theta}_1$ is distributed according to the target $p_{\Theta|Y}$. The trajectory is obtained as a solution of the following ODE:

$$\frac{\mathrm{d}\boldsymbol{\theta}_t}{\mathrm{d}t} = u_{\Theta}\left(t, \boldsymbol{\theta}_t, \boldsymbol{y}\right), \quad \forall t \in [0,1], \qquad \text{with } \boldsymbol{\theta}_0 = \boldsymbol{\theta}.$$

The time-dependent flow $\psi_{\Theta}$ associated to the above ODE is simply given by samples along the path starting from the initial condition $\boldsymbol{\theta}$, i.e. $\psi_{\Theta}(t, \boldsymbol{\theta}, \boldsymbol{y}) := \boldsymbol{\theta}_t$. Such a flow allows defining the transport map from $\pi_{\Theta}$ to $p_{\Theta|Y}$ as $T_{\Theta}(\boldsymbol{\theta}, \boldsymbol{y}) := \psi_{\Theta}(1, \boldsymbol{\theta}, \boldsymbol{y})$. The vector field is approximated with a deep neural network $\hat{u}_{\Theta}$ trained on a *guided* version of the conditional flow-matching loss from Lipman et al. (2023) with a linear interpolation strategy:

$$\mathcal{L}_{\Theta}(\hat{u}_{\Theta}) = \mathbb{E}\left[\int_0^1 \|\hat{u}_{\Theta}(t, \boldsymbol{\theta}_t, \boldsymbol{y}) - (\boldsymbol{\theta}_1 - \boldsymbol{\theta}_0)\|^2 \, \mathrm{d}t\right], \quad \boldsymbol{\theta}_t := (1-t)\boldsymbol{\theta}_0 + t\boldsymbol{\theta}_1 \; \forall t \in [0,1] \,, \tag{5}$$

where the expectation is taken over $(\boldsymbol{\theta}_1, \boldsymbol{y}, \boldsymbol{\theta}_0) \sim p_{\Theta,Y}(\boldsymbol{\theta}_1, \boldsymbol{y})\pi_{\Theta}(\boldsymbol{\theta}_0)$. Here, the high-fidelity dataset $\mathcal{D}_{\text{cal}}$ can be used to provide joint samples $(\boldsymbol{\theta}_1, \boldsymbol{y})$ from $p_{\Theta,Y}(\boldsymbol{\theta}_1, \boldsymbol{y})$, thus providing an empirical version of the above objective.

In most flow matching instances, the base distribution $\pi_{\Theta}$ is set to a simple standardized Gaussian distribution, i.e. $\pi_{\Theta}(\boldsymbol{\theta}) = \mathcal{N}(\boldsymbol{\theta} \mid \mathbf{0}, \boldsymbol{I}_p)$. However, for settings such as the one we consider here, where $\mathcal{D}_{\text{cal}}$ is small, this choice may result in very poor posterior approximators with high variance. Instead, we use the simulator to construct a more informative data-driven source distribution $\pi_{\Theta} = \pi_{\Theta|Y}$ that acts as a possibly low-quality surrogate of the true $p_{\Theta|Y}$. Training itself remains consistent with standard guided flow matching; the learned flow then serves to refine this source distribution using only the limited high-fidelity data.

### 3.2 SIMULATION-INFORMED SOURCE DISTRIBUTION

The source distribution $\pi_{\Theta|Y}$ plays a central role in our framework. Ideally, it should be very close to the true posterior distribution, so that the flow $\psi_{\Theta}$ induces minimal corrections to $\boldsymbol{\theta}_0 \sim \pi_{\Theta|Y}$. Making use of the availability of the simulator $S$, a first natural possibility is to set $\pi_{\Theta|Y}(\boldsymbol{\theta}|\boldsymbol{y}) = \hat{p}_{\Theta|X}(\boldsymbol{\theta}|\boldsymbol{y})$ where $\hat{p}_{\Theta|X}$ is a posterior approximation obtained via SBI. This provides a reasonable approximation of the true $p_{\Theta|Y}$ in the absence of misspecification but is likely to be a poor one in less favorable settings. A natural alternative is to plug into $\hat{p}_{\Theta|X}$ a conditioning sample from a

distribution $q_{\boldsymbol{X}|\boldsymbol{Y}}(\boldsymbol{x}|\boldsymbol{y})$ with the same support as $p_{\boldsymbol{X}}(\boldsymbol{x})$ and informative of $\boldsymbol{y}$. This corresponds to considering a source distribution of the form

$$\pi_{\boldsymbol{\Theta}|\boldsymbol{Y}}(\boldsymbol{\theta}|\boldsymbol{y}) = \int \hat{p}_{\boldsymbol{\Theta}|\boldsymbol{X}}(\boldsymbol{\theta}|\boldsymbol{x})\, q_{\boldsymbol{X}|\boldsymbol{Y}}(\boldsymbol{x}|\boldsymbol{y})\, d\boldsymbol{x}. \tag{6}$$

The most straightforward choice for $q_{\boldsymbol{X}|\boldsymbol{Y}}$ is $q_{\boldsymbol{X}|\boldsymbol{Y}}(\boldsymbol{x}|\boldsymbol{y}) := p_{\boldsymbol{X}|\boldsymbol{Y}}(\boldsymbol{x}|\boldsymbol{y})$, which would require, in addition to $\mathcal{D}_{\text{cal}}$, a large number of ground truth samples $(\boldsymbol{y}, \boldsymbol{x})$ for training an approximator. Thus, we consider instead,

$$q_{\boldsymbol{X}|\boldsymbol{Y}}(\boldsymbol{x}|\boldsymbol{y}) := \int p_{\boldsymbol{X}|\boldsymbol{\Theta}}(\boldsymbol{x}|\boldsymbol{\theta})\, p_{\boldsymbol{\Theta}|\boldsymbol{Y}}(\boldsymbol{\theta}|\boldsymbol{y})\, d\boldsymbol{\theta}, \tag{7}$$

that can be seen as a mixture and approximated using the simulator $S$ and joint calibration pairs $(\boldsymbol{\theta}_j, \boldsymbol{y}_j) \sim p_{\boldsymbol{\Theta}, \boldsymbol{Y}}(\boldsymbol{\theta}, \boldsymbol{y}) = p_{\boldsymbol{\Theta}|\boldsymbol{Y}}(\boldsymbol{\theta}|\boldsymbol{y})p_{\boldsymbol{Y}}(\boldsymbol{y})$. Indeed, note that for each $\boldsymbol{y}_j \in \mathcal{D}_{\text{cal}}$, its associated $\boldsymbol{\theta}_j$ is a sample from $p_{\boldsymbol{\Theta}|\boldsymbol{Y}}(\boldsymbol{\theta}|\boldsymbol{y}_j)$ that can be plugged in the simulator to generate $\boldsymbol{x}_j \sim p_{\boldsymbol{X}|\boldsymbol{\Theta}}(\boldsymbol{x}|\boldsymbol{\theta}_j)$, leading to $\boldsymbol{x}_j \sim q_{\boldsymbol{X}|\boldsymbol{Y}}(\boldsymbol{x}|\boldsymbol{y}_j)$. This is typically used in lines 2 and 3 of Algorithm 2.

In practice, a sample $\boldsymbol{\theta}$ from the source distribution (6), can be obtained by first drawing a sample $\tilde{\boldsymbol{x}}$ from $q_{\boldsymbol{X}|\boldsymbol{Y}}(\tilde{\boldsymbol{x}}|\boldsymbol{y})$ in (7), then setting $\boldsymbol{\theta}$ to a sample from $\hat{p}_{\boldsymbol{\Theta}|\boldsymbol{X}}(\boldsymbol{\theta}|\tilde{\boldsymbol{x}})$. As we can already easily sample from $\hat{p}_{\boldsymbol{\Theta}|\boldsymbol{X}}$, it only remains to provide a way to sample from $q_{\boldsymbol{X}|\boldsymbol{Y}}(\boldsymbol{x}|\boldsymbol{y})$ for any $\boldsymbol{y}$. To this end, we use the flow matching framework to construct a conditional transport map $T_{\boldsymbol{X}} : \mathbb{R}^d \times \mathbb{R}^d \to \mathbb{R}^d$, from a Gaussian distribution centered around $\boldsymbol{y}$ with isotropic variance $\sigma^2$, towards $q_{\boldsymbol{X}|\boldsymbol{Y}}(\boldsymbol{x}|\boldsymbol{y})$ i.e.:

$$\boldsymbol{x}_0 \sim \mathcal{N}(\boldsymbol{x}|\boldsymbol{y}, \sigma^2 \boldsymbol{I}_d) \to \tilde{\boldsymbol{x}} = T_{\boldsymbol{X}}(\boldsymbol{x}_0, \boldsymbol{y}) \approx q_{\boldsymbol{X}|\boldsymbol{Y}}(\boldsymbol{x}|\boldsymbol{y}). \tag{8}$$

More specifically, we define $T_{\boldsymbol{X}}(\boldsymbol{x}_0, \boldsymbol{y}) = \psi_{\boldsymbol{X}}(1, \boldsymbol{x}_0, \boldsymbol{y})$, where $\psi_{\boldsymbol{X}}$ is the flow associated to the velocity field neural approximator $\hat{u}_{\boldsymbol{X}}$ learned by minimizing the following objective:

$$\mathcal{L}_{\boldsymbol{X}}(\hat{u}_{\boldsymbol{X}}) = \mathbb{E}\left[\int_0^1 \|\hat{u}_{\boldsymbol{X}}(t, \boldsymbol{x}_t, \boldsymbol{y}) - (\boldsymbol{x}_1 - \boldsymbol{x}_0)\|^2\, dt\right], \; \boldsymbol{x}_t := (1-t)\boldsymbol{x}_0 + t\boldsymbol{x}_1\; \forall t \in [0, 1], \tag{9}$$

where the expectation is taken over $(\boldsymbol{y}, \boldsymbol{x}_1, \boldsymbol{x}_0) \sim q_{\boldsymbol{Y}, \boldsymbol{X}}(\boldsymbol{y}, \boldsymbol{x}_1)\mathcal{N}(\boldsymbol{x}_0|\boldsymbol{y}, \sigma^2 \boldsymbol{I}_d)$. Here, the dataset $\mathcal{D}_{\text{cal}}$ and simulator $S$ can be used to provide joint samples $(\boldsymbol{y}, \boldsymbol{x}_1)$ from $q_{\boldsymbol{Y}, \boldsymbol{X}}(\boldsymbol{y}, \boldsymbol{x}_1)$ as discussed later in Section 3.3. We took inspiration from Albergo et al. (2024) to define a source distribution that induces a coupling between the base and target distributions through the conditioning variable, which greatly helps the training process under limited data.

When the conditional independence hypothesis in Equation (2) is valid and chosing $q_{\boldsymbol{X}|\boldsymbol{Y}}(\boldsymbol{x}|\boldsymbol{y}) = p_{\boldsymbol{X}|\boldsymbol{Y}}(\boldsymbol{x}|\boldsymbol{y})$, we can use Equation (3) to see that $p_{\boldsymbol{X}|\boldsymbol{Y}}(\boldsymbol{x}|\boldsymbol{y})$ is indeed the optimal choice for transporting data from $\boldsymbol{y}$-space to $\boldsymbol{x}$-space, since

$$p_{\boldsymbol{\Theta}|\boldsymbol{Y}}(\boldsymbol{\theta}|\boldsymbol{y}) = \mathbb{E}_{\boldsymbol{X}|\boldsymbol{Y}}\left[p_{\boldsymbol{\Theta}|\boldsymbol{X}}(\boldsymbol{\theta}|\boldsymbol{x})|\boldsymbol{y}\right],$$
$$\tilde{\boldsymbol{x}} \sim p_{\boldsymbol{X}|\boldsymbol{Y}}(\boldsymbol{x}|\boldsymbol{y}) \text{ and } \boldsymbol{\theta} \sim p_{\boldsymbol{\Theta}|\boldsymbol{X}}(\boldsymbol{\theta}|\tilde{\boldsymbol{x}}) \implies \boldsymbol{\theta} \sim p_{\boldsymbol{\Theta}|\boldsymbol{Y}}(\boldsymbol{\theta}|\boldsymbol{y}),.$$

In that case, learning only $\hat{u}_{\boldsymbol{X}}$ could be seen as sufficient, since the proposal $\pi_{\boldsymbol{\Theta}|\boldsymbol{Y}}$ is already a good approximation to the true posterior. However, if the conditional independence is no longer valid, if $\hat{p}_{\boldsymbol{\Theta}|\boldsymbol{X}}$ is poorly trained, or $\hat{u}_{\boldsymbol{X}}$ is not optimal, the proposal would not have the flexibility to compensate for errors from each of its different parts. This is no longer an issue when using $\hat{u}_{\boldsymbol{X}}$ to define a source distribution for learning the vector field $\hat{u}_{\boldsymbol{\Theta}}$, as proposed in this work, since it does not rely on the validity of Equation (2).

### 3.3 JOINT TRAINING OF POSTERIOR AND SOURCE DISTRIBUTIONS BY FLOW MATCHING

We now face two optimization tasks: minimizing the loss in (9) to train the simulation-space vector field $\hat{u}_{\boldsymbol{X}}$ and minimizing (5) to train the parameter-space vector field $\hat{u}_{\boldsymbol{\Theta}}$. Instead of solving them separately, we propose to optimize the following *joint objective* for improved practical performance:

$$\mathcal{L}_{\boldsymbol{\Theta}, \boldsymbol{X}}(\hat{u}_{\boldsymbol{X}}, \hat{u}_{\boldsymbol{\Theta}}) = \mathbb{E}\left[\int_0^1 \|\hat{u}_{\boldsymbol{\Theta}}(t, \boldsymbol{\theta}_t, \boldsymbol{y}) - (\boldsymbol{\theta}_1 - \boldsymbol{\theta}_0)\|^2 + \|\hat{u}_{\boldsymbol{X}}(t, \boldsymbol{x}_t, \boldsymbol{y}) - (\boldsymbol{x}_1 - \boldsymbol{x}_0)\|^2\, dt\right],$$

with $(\boldsymbol{x}_1, \boldsymbol{\theta}_1, \boldsymbol{y}, \boldsymbol{x}_0, \boldsymbol{\theta}_0) \sim q_{\boldsymbol{X}|\boldsymbol{Y}}(\boldsymbol{x}_1|\boldsymbol{y})p_{\boldsymbol{\Theta}, \boldsymbol{Y}}(\boldsymbol{\theta}_1, \boldsymbol{y})\pi_{\boldsymbol{\Theta}|\boldsymbol{Y}}(\boldsymbol{\theta}_0|\boldsymbol{y})\mathcal{N}(\boldsymbol{x}_0, \boldsymbol{y}, \sigma^2 I_d)$ and where $\boldsymbol{\theta}_t$ and $\boldsymbol{x}_t$ are convex combinations as in Equations (5) and (9). An important specificity of the above

objective compared to standard flow matching is that the source sample $\boldsymbol{\theta}_0$ depends on $\hat{u}_{\boldsymbol{X}}$, by definition of the source $\pi_{\boldsymbol{\Theta}|\boldsymbol{Y}}$, which is evolving during training. Intuitively, this joint formulation forces $\hat{u}_{\boldsymbol{\Theta}}$ to be robust to noisy or inaccurate samples from the source distribution: during early training stages, $\pi_{\boldsymbol{\Theta}|\boldsymbol{Y}}$ may yield poor candidates $\boldsymbol{\theta}_0$, yet $\hat{u}_{\boldsymbol{\Theta}}$ must learn to accommodate them. This robustness is desirable at test time, where the source distribution may also generate imperfect samples for previously unseen observations $\boldsymbol{y}$.

The joint learning of vector fields $\hat{u}_{\boldsymbol{X}}$ and $\hat{u}_{\boldsymbol{\Theta}}$ is summarized in Algorithm 1, which specifies how to optimize the objective $\mathcal{L}_{\boldsymbol{\Theta},\boldsymbol{X}}(\hat{u}_{\boldsymbol{X}}, \hat{u}_{\boldsymbol{\Theta}})$ using the sampling procedure from Algorithm 2 referred to as function SAMPLETRAININGTUPLE. In practice, variable $t$ is sampled i.i.d. for each of the terms $\boldsymbol{X}$ and $\boldsymbol{\Theta}$ from $\mathcal{L}_{\boldsymbol{X},\boldsymbol{\Theta}}$ (lines 5 and 6) to reduce bias. Also, because $\hat{u}_{\boldsymbol{X}}$ affects the distribution of $\boldsymbol{\theta}_0$ (via $\tilde{\boldsymbol{x}}$), the effective source distribution for $\hat{u}_{\boldsymbol{\Theta}}$ is non-stationary; to mitigate instability we use small learning rates and gradient clipping to reduce training instability in the early stages.

---

**Algorithm 1** Joint flow training for FMCPE

---

**Require:** SAMPLETRAININGTUPLE function, trainable flows $\hat{u}_{\boldsymbol{X}}, \hat{u}_{\boldsymbol{\Theta}}$ and minibatch size $B$.

1: **repeat**
2:      $\mathcal{L} \leftarrow 0$
3:      **for** $i = 1$ **to** $B$ **do**
4:          $(\boldsymbol{y}, \boldsymbol{\theta}_1, \boldsymbol{\theta}_0, \boldsymbol{x}_1, \boldsymbol{x}_0) \leftarrow$ SAMPLETRAININGTUPLE$(\hat{u}_{\boldsymbol{X}})$             ▷ see Algorithm 2
5:          Draw $t \sim \mathcal{U}[0,1]$ and set $\boldsymbol{x}_t = (1-t)\boldsymbol{x}_0 + t\,\boldsymbol{x}_1$
6:          Draw *another independent* $\tau \sim \mathcal{U}[0,1]$ and set $\boldsymbol{\theta}_\tau = (1-\tau)\boldsymbol{\theta}_0 + \tau\,\boldsymbol{\theta}_1$
7:          $\ell_{\boldsymbol{X}} \leftarrow \|\hat{u}_{\boldsymbol{X}}(t, \boldsymbol{x}_t, \boldsymbol{y}) - (\boldsymbol{x}_1 - \boldsymbol{x}_0)\|^2$
8:          $\ell_{\boldsymbol{\Theta}} \leftarrow \|\hat{u}_{\boldsymbol{\Theta}}(\tau, \boldsymbol{\theta}_\tau, \boldsymbol{y}) - (\boldsymbol{\theta}_1 - \boldsymbol{\theta}_0)\|^2$
9:          $\mathcal{L} \leftarrow \mathcal{L} + (\ell_{\boldsymbol{X}} + \ell_{\boldsymbol{\Theta}})$
10:     **end for**
11:     $\mathcal{L} \leftarrow \mathcal{L}/B$
12:     Take one optimizer step on trainable parameters of $\hat{u}_{\boldsymbol{X}}$ and $\hat{u}_{\boldsymbol{\Theta}}$ using $\mathcal{L}$
13: **until** convergence

---

Algorithm 2 first samples $(\boldsymbol{\theta}_1, \boldsymbol{y})$ from the calibration dataset, where $\boldsymbol{\theta}_1$ serves as target sample to the vector field $\hat{u}_{\boldsymbol{\Theta}}$. We then generate $\boldsymbol{x}_1 \sim p_{\boldsymbol{X}|\boldsymbol{\Theta}}(\boldsymbol{x}|\boldsymbol{\theta}_1)$ which serves as target for $\hat{u}_{\boldsymbol{X}}$ and is approximately distributed according to $q_{\boldsymbol{X}|\boldsymbol{Y}}(\boldsymbol{x}|\boldsymbol{y})$ (see Equation (7)). The source sample $\boldsymbol{x}_0$ is obtained from a Gaussian distribution centered at $\boldsymbol{y}$. To obtain a source sample $\boldsymbol{\theta}_0$ for the vector field $\hat{u}_{\boldsymbol{\Theta}}$, an intermediate sample $\tilde{\boldsymbol{x}}$ is computed by solving an ODE driven by the current estimated vector field $\hat{u}_{\boldsymbol{X}}$ and starting from $\boldsymbol{x}_0$. This sample is simply provided to the approximate posterior to get $\boldsymbol{\theta}_0 \sim \hat{p}_{\boldsymbol{\Theta}|\boldsymbol{X}}(\boldsymbol{\theta}|\tilde{\boldsymbol{x}})$ as discussed in Section 3.2. Note that one could alternatively generate $\tilde{\boldsymbol{x}} = S(\boldsymbol{\theta}_1, \epsilon)$ directly from the simulator instead of solving the ODE induced by $\hat{u}_{\boldsymbol{X}}$. However, this strategy cannot be applied at inference time, where only the observation $\boldsymbol{y}$ is available and not the pair $(\boldsymbol{\theta}, \boldsymbol{y})$. For consistency between training and inference, we therefore sample $\tilde{\boldsymbol{x}}$ using the ODE. A crucial aspect in the above sampling procedure is to prevent gradients from propagating through the intermediate sample $\tilde{\boldsymbol{x}}$ when optimizing the vector field $\hat{u}_{\boldsymbol{X}}$ which would bias its training.

---

**Algorithm 2** SAMPLETRAININGTUPLE$(\hat{u}_{\boldsymbol{X}})$

---

**Require:** Calibration set $\mathcal{D}_{\mathrm{cal}}$, simulator $S$, pretrained SBI model $\hat{p}_{\boldsymbol{\Theta}|\boldsymbol{X}}(\boldsymbol{\theta} \mid \tilde{\boldsymbol{x}})$ (frozen) and velocity field $\hat{u}_{\boldsymbol{X}}$

1: **function** SAMPLETRAININGTUPLE$(\hat{u}_{\boldsymbol{X}})$
2:     Sample $(\boldsymbol{\theta}_1, \boldsymbol{y}) \sim \mathcal{D}_{\mathrm{cal}}$                                 ▷ calibration samples
3:     Sample $\boldsymbol{x}_1$ using simulator $S$ evaluated at $\boldsymbol{\theta}_1$                ▷ see Equation (7)
4:     Draw base sample $\boldsymbol{x}_0 \sim \mathcal{N}(\boldsymbol{x}_0, \boldsymbol{y}, \sigma^2 I)$                   ▷ base for $\boldsymbol{X}$-flow
5:     Solve ODE: $\frac{d\bar{\boldsymbol{x}}_t}{dt} = \hat{u}_{\boldsymbol{X}}(t, \bar{\boldsymbol{x}}_t, \boldsymbol{y})$ with $\bar{\boldsymbol{x}}_0 = \boldsymbol{x}_0$.
6:     Set $\tilde{\boldsymbol{x}} \leftarrow T_{\boldsymbol{X}}(\boldsymbol{x}_0; \boldsymbol{y}) := \bar{\boldsymbol{x}}_1$          ▷ map base → simulator-space (uses current $\hat{u}_{\boldsymbol{X}}$)
7:     Set $\tilde{\boldsymbol{x}} \leftarrow$ StopGradient$(\tilde{\boldsymbol{x}})$             ▷ do not propagate gradient through sample
8:     Sample $\boldsymbol{\theta}_0 \sim \hat{p}_{\boldsymbol{\Theta}|\boldsymbol{X}}(\boldsymbol{\theta}|\tilde{\boldsymbol{x}})$                             ▷ sample from source
9:     **return** $(\boldsymbol{y}, \boldsymbol{\theta}_1, \boldsymbol{\theta}_0, \boldsymbol{x}_1, \boldsymbol{x}_0)$
10: **end function**

---

## 4 EXPERIMENTS

We benchmark our method against three baselines, NPE, MFNPE, and RoPE. The comparison is carried out on four tasks described in Section 4.1, using various evaluation metrics specified in Section 4.2. We follow the implementation from Wehenkel et al. (2025) for NPE and train it using only the limited calibration data $\mathcal{D}_{\text{cal}}$. Note that since the simulator is not used, we expect this baseline NPE to fail as the complexity of the generating process increases. For each experiment, all evaluations are performed on a test set $\mathcal{D}_{\text{test}} = \{(\boldsymbol{\theta}_j, \boldsymbol{y}_j)\}_{1 \leq j \leq N_{\text{test}}}$ with $N_{\text{test}} = 2000$ unless otherwise specified, and each metric is reported for different sizes $N_{\text{cal}}$ of the calibration set. Calibration sets are constructed in an expanding manner, gradually adding new samples to an initial set, to limit the sources of variability in the comparison; see Appendix A for details.

All experiments described below are implemented in Python using pytorch (Paszke et al., 2019) and mlxp (Arbel & Zouaoui, 2024). We also use nflows (Durkan et al., 2020) and the dingo (Dax et al., 2021) packages for the implementation of continuous normalizing flows. Our code is provided as a zip file in the supplementary materials.

### 4.1 TASKS

Our experiments consist of two synthetic and two real-world tasks. Setup and implementation details can be found in Appendix A.

▶ Gaussian: A multivariate Gaussian model is considered, with $\boldsymbol{\theta} \in \mathbb{R}^3$ and $\boldsymbol{\theta} \sim \mathcal{N}(\mu_\theta, \Sigma_\theta)$. Both $\boldsymbol{x} \in \mathbb{R}^{10}$ and $\boldsymbol{y} \in \mathbb{R}^{10}$ follow multivariate Gaussian distributions centered on different linear combinations of $\boldsymbol{\theta}$.

▶ Pendulum: The damped pendulum (Takeishi & Kalousis, 2021) models the oscillations of a mass around a fixed attachment point. Parameters $\boldsymbol{\theta} = [A, \omega_0]$ are the oscillation amplitude $A$ and the natural frequency $\omega_0$. Simulations are generated from a simplified model that omits friction forces, thus creating a systematic misspecification relative to the real dynamics. Both observations and simulations are real-valued time-series of length 200.

▶ WindTunnel: This task from Gamella et al. (2025) consists of measuring the air pressure inside an horizontal tube where air is being pushed through by two controllable fans at both ends. The goal is to infer the opening angle (in degrees) of a hatch $H \in [0, 45]$ on the side, given the pressure values inside the tunnel after applying a short power impulse to the intake fan. For the simulator, we use the model A2C3 from (Gamella et al., 2025, Appendix IV).

▶ LightTunnel: In this task from Gamella et al. (2025), a camera is capturing light passing through two linear polarizers inside an elongated chamber. The goal is to predict RGB values of the light source and the polarizer effect $\alpha \in [0, 1]$, which is a function of the polarizer angle, so that $\boldsymbol{\theta} := [R, G, B, \alpha] \in [0, 255]^3 \times [0, 1]$. The simulator is a simplification of the real world process and described in (Gamella et al., 2025, Appendix IV) (Model F1). Observations are RGB images of size $(W, H, C) = (64, 64, 3)$ produced by either the simulator or the real apparatus; model misspecification arises because the simulator omits certain physical effects present in the real measurements.

### 4.2 EVALUATION METRICS

We assess the performance of our method performance based on four metrics.

**Classifier Two-Sample Test (C2ST) and joint C2ST ($j$C2ST).** The C2ST (Lopez-Paz & Oquab, 2017) measures the discrepancy between two distributions by training a binary classifier to distinguish samples drawn from them. The test statistic is the classifier's accuracy on samples from a test set, which is equal to chance-level (usually 0.5) for indistinguishable distributions and approaches 1 as the distributions diverge. This is the metric used for the Gaussian example. For the other tasks, since our reference $\mathcal{D}_{\text{test}}$ provides a single ground truth value $\boldsymbol{\theta}_j$ for each $\boldsymbol{y}_j$, standard C2ST cannot be used, as it would require multiple samples from the true posterior $p(\boldsymbol{\theta}|\boldsymbol{y}_j)$. Instead, for each $\boldsymbol{y}_j$ we generate one $\tilde{\boldsymbol{\theta}}_j$ from the learned posterior and compare $\mathcal{D}_{\text{test}}$ to the set of pairs $\{(\tilde{\boldsymbol{\theta}}_j, \boldsymbol{y}_j)\}_{1 \leq j \leq N_{\text{test}}}$ as samples from a joint distribution, justifying the name $j$C2ST.

**Wasserstein Distance** ($W_2$). Similarly, the Wasserstein distance for the $L_2$ cost is computed between samples from the approximate and true joint distribution $p(\boldsymbol{\theta}, \boldsymbol{y})$ to provide a geometry-aware notion of distributional similarity.

$$W_2 = \min_{\gamma \in \mathbb{R}_+^{N_{\text{test}} \times N_{\text{test}}}} \left( \sum_{i,j} \gamma_{i,j} \|(\boldsymbol{\theta}_i, \boldsymbol{y}_i) - (\tilde{\boldsymbol{\theta}}_j, \boldsymbol{y}_j)\|_2^2 \right)^{\frac{1}{2}} \quad \text{such that } \gamma.\mathbf{1} = \frac{1}{N_{\text{test}}} \text{ and } \gamma^T.\mathbf{1} = \frac{1}{N_{\text{test}}} .$$

For the Gaussian task, the $W_2$ distance is computed between parameter samples only since the ground truth posterior distribution is available.

**Maximum Mean Discrepancy** (MMD). Analogously, we compute the Maximum Mean Discrepancy between samples from the true and approximate joint distributions using a radial basis function (RBF) kernel. Given samples $\{(\boldsymbol{\theta}_i, \boldsymbol{y}_i)\}_{1 \leq i \leq N_{\text{test}}}$ and $\{(\tilde{\boldsymbol{\theta}}_j, \boldsymbol{y}_j)\}_{1 \leq j \leq N_{\text{test}}}$, the squared MMD is

$$\text{MMD}^2 = \frac{1}{N_{\text{test}}(N_{\text{test}} - 1)} \sum_{i \neq i'} k_\sigma((\boldsymbol{\theta}_i, \boldsymbol{y}_i), (\boldsymbol{\theta}_{i'}, \boldsymbol{y}_{i'})) + \frac{1}{N_{\text{test}}(N_{\text{test}} - 1)} \sum_{j \neq j'} k_\sigma\left((\tilde{\boldsymbol{\theta}}_j, \boldsymbol{y}_j), (\tilde{\boldsymbol{\theta}}_{j'}, \boldsymbol{y}_{j'})\right)$$

$$- \frac{2}{N_{\text{test}}^2} \sum_{i,j} k_\sigma\left((\boldsymbol{\theta}_i, \boldsymbol{y}_i), (\tilde{\boldsymbol{\theta}}_j, \boldsymbol{y}_j)\right), \quad k_\sigma(x, x') = \exp\left(-\frac{\|x - x'\|_2^2}{2\sigma^2}\right)$$

with $\sigma = 10$ in our experiments.

**Mean Squared Error (MSE)**. We also report the average mean squared error between $M$ generated samples $\{\tilde{\boldsymbol{\theta}}_j^i\}_{1 \leq i \leq M}$ and the ground truth parameters $\{\boldsymbol{\theta}_j\}_{1 \leq j \leq N_{\text{test}}}$ for each observation $\boldsymbol{y}_j$ in $\mathcal{D}_{\text{test}}$.

$$\text{MSE} = \frac{1}{N_{\text{test}}} \frac{1}{M} \sum_j^{N_{\text{test}}} \sum_i^M \|\tilde{\boldsymbol{\theta}}_j^i - \boldsymbol{\theta}_i\|_2^2 .$$

MSE is a good accuracy measure for unimodal posteriors, which is often the case in our experiments.

### 4.3 RESULTS AND DISCUSSION

Figure 2 and Figure 6 (in Appendix), first illustrate the impact of $N_{\text{cal}}$ on posterior estimation. All metrics decrease as $N_{\text{cal}}$ increases, highlighting the importance of having enough calibration data to correct misspecification. Note that for `WindTunnel` we sometimes observe $j$C2ST values below 0.5. This is due to an insufficient test set with respect to the dimensions of the problem, causing the classifier used for $j$C2ST to not be sufficiently well trained ($N_{\text{test}}$ was set to 5000 for this task, the maximal number of validation data points provided by the `causal-chamber` package). For a given ground truth couple $(\boldsymbol{\theta}^*, \boldsymbol{y}^*)$, Figure 3 illustrates the posterior density of the first two components of $\boldsymbol{\theta}$ for each method (except `RoPE`), for `Gaussian` and `Pendulum`. Similar plots for the other tasks are provided in the Appendix in Figures 7 (`WindTunnel`) and 8 (`LightTunnel`).

All experiments demonstrate that our method consistently outperforms all baselines, quantitatively and qualitatively. In Figure 2, we observe that even on the simple `Gaussian` task, our approach achieves better performance than plain NPE, fine-tuned `MFNPE`, and `RoPE`. This gap becomes more striking in Figure 3 (second row), where both NPE and `MFNPE` produce multimodal posteriors even when the true posterior is unimodal. In contrast, our method yields unimodal posteriors that are better calibrated and centered on the ground-truth parameter $\boldsymbol{\theta}^*$, and this even for a low calibration size. For more complex tasks, such as `Pendulum` and `LightTunnel`, NPE fails to capture the intricate dependencies between parameters and observations—even when $N_{\text{cal}} = 1000$. In these cases, our method achieves superior performance in both $j$C2ST and $W_2$, while also exhibiting lower variance across calibration seeds compared to `MFNPE` (Figure 2). Figure 3 further illustrates that `MFNPE` often produces overly sharp posterior distributions in one or more dimensions, yet fails to recover the true parameter $\boldsymbol{\theta}^*$. In contrast, our method consistently recovers $\boldsymbol{\theta}^*$ with higher accuracy. We hypothesize that this behavior stems from the `MFNPE` training procedure. `MFNPE` learns a neural encoder $h_\omega(\boldsymbol{x})$ to extract latent representations from simulated data $\boldsymbol{x}$. During fine-tuning, however, this encoder is evaluated on real observations $\boldsymbol{y}$, whose distribution differs from that of $\boldsymbol{x}$. This distributional shift leads to erroneous latent representations $h_\omega(\boldsymbol{y})$, and consequently to biased posteriors. While this issue diminishes as $N_{\text{cal}}$ increases, it remains pronounced at small

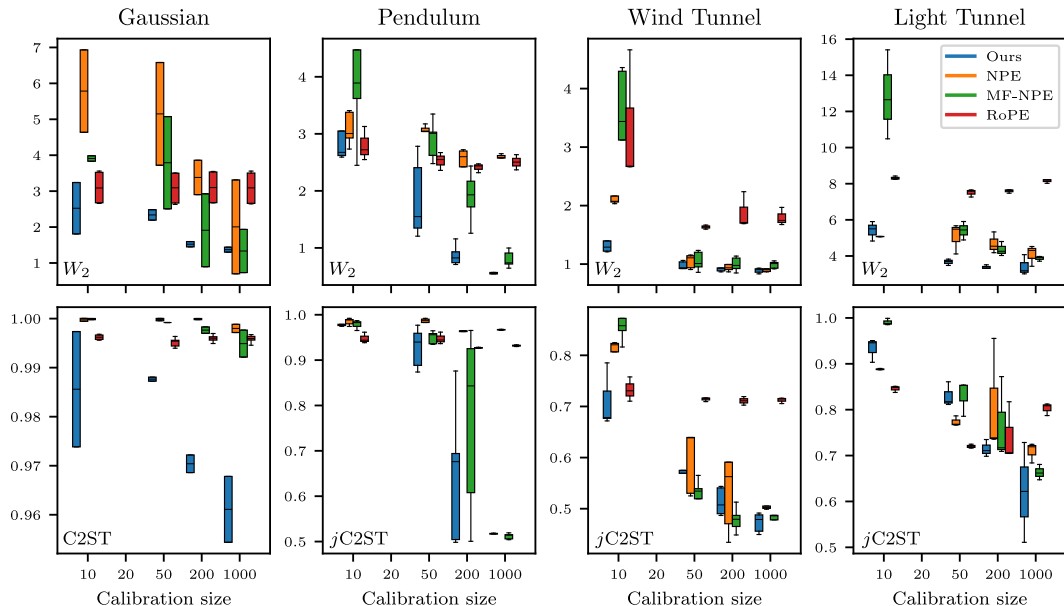

Figure 2: Wasserstein distance (top row, ↓ is better) and C2ST/$j$C2ST (bottom row, ↓ is better) with respect to an increasing calibration set size $N_{\text{cal}} \in \{10, 50, 200, 1000\}$. Each boxplot shows the distribution of metric values across five independent runs, each using a different randomly chosen calibration set. Note that for the Gaussian Task (left), the C2ST and $W_2$ distance are averaged over 3 different observations chosen at random.

calibration sizes ($N_{\text{cal}} = 10$ or $50$). The `Pendulum` task provides further evidence supporting this analysis. The misspecification arises from exponential damping, which predominantly affects the estimation of the oscillation amplitude $A$, as seen in Figure 3 (first row). In contrast, the estimation of the frequency $\omega_0$ remains accurate, since it is less sensitive to the damping mismatch. In the real-world tasks (`WindTunnel` and `LightTunnel`), our approach yields substantially better results at small calibration sizes, particularly in terms of $W_2$, and remains competitive or superior for larger calibration sets. We provide additional visualizations of the posterior estimates for these tasks in Appendix B.

To provide more insight on the different `FMCPE` components, an ablation study is also conducted. Our joint `FMCPE` training, referred to as $T_{\Theta} + T_{X}$ is first compared to a sequential one where the $T_{X}$ map is first learned to provide the source distribution for $T_{\Theta}$ fitted in turns. Figure 4 shows that in all tested examples and metrics, the joint training provides better or equivalent performance, confirming the advantage of adjusting both maps in an interactive manner. To explore then the impact of each map separately, two options are considered. In the $T_{X}$-only procedure, NPE is used with $\hat{p}_{\Theta|X}(\theta|X = T_{X}(y))$ where $T_{X}$ is trained on calibration data, while $T_{\Theta}$-only denotes a learned $T_{\Theta}$ on calibration data from a source distribution $\pi(\theta|y) = \hat{p}_{\Theta|X}(\theta|X = y)$ ignoring calibration data. The worse performance of $T_{X}$-only, including when calibration data increases, suggests that calibration has more impact when learning $T_{\Theta}$ than $T_{X}$. In contrast, $T_{\Theta}$-only and $T_{\Theta} + T_{X}$ − Sequential, which differ mainly in the nature of the source, show similar performance. This suggests that a key feature of `FMCPE` is the possibility to learn jointly the source and $T_{\Theta}$ rather than the nature of the source itself. Note that the `Gaussian` example is not shown because no significant differences were observed, which suggests that for simple models a sub-optimal use of calibration data may not be critical.

## 5 CONCLUSION

We tackled model misspecification in SBI by combining scarce calibration data with abundant simulations. Our method builds a proposal posterior from both sources and refines it via flow matching, producing posterior estimates that are more accurate and better calibrated than standard SBI baselines. Importantly, our proposal is also computationally efficient, as it can leverage off-the-shelf SBI posterior distributions as proposals, requiring only lightweight refinement with calibration data.

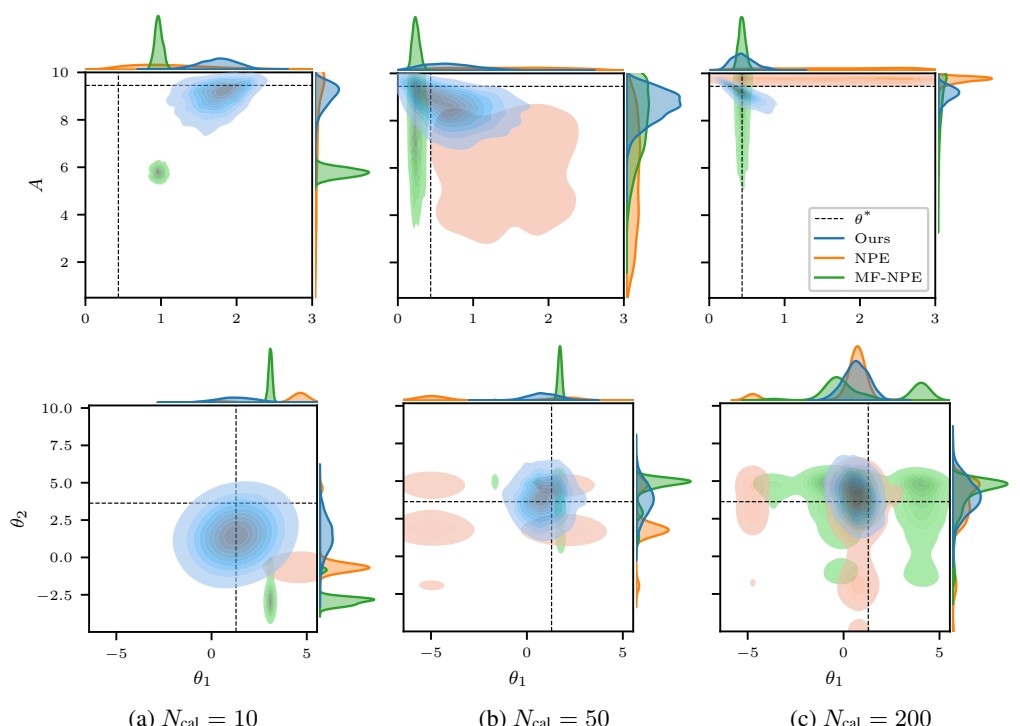

Figure 3: Kernel density estimates of joint and marginal samples for Tasks B (first row) and A (second row). For a given $\boldsymbol{y}^* \in \mathcal{D}_{\text{test}}$, we draw $\{\tilde{\boldsymbol{\theta}}_i\}_{1 \leq i \leq 2000}$, for each method and 3 calibration sizes $N_{\text{cal}} \in \{10, 50, 200\}$. Dotted black lines indicate the true parameter $\boldsymbol{\theta}^*$ that generated $\boldsymbol{y}^*$.

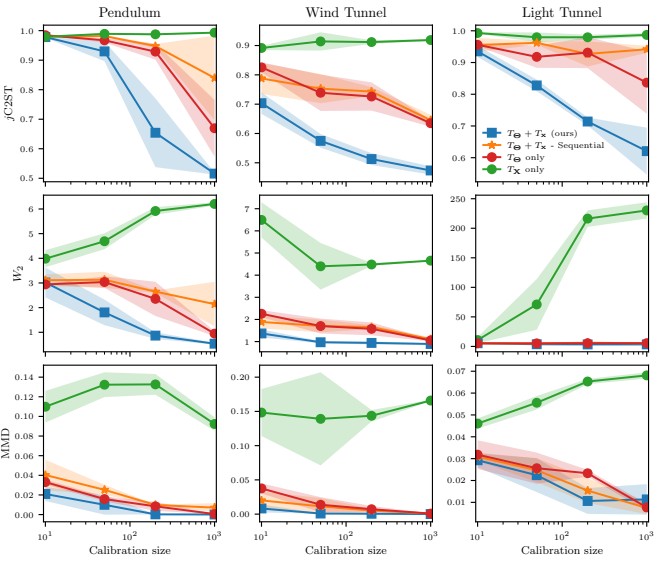

Figure 4: Ablation study investigating the impact of each `FMCPE` component (the lower the better). Our joint $T_\Theta + T_X$ (blue) is compared to a sequential version (orange) learning $T_X$ then $T_\Theta$, and to two other options. In $T_X$ only (green), resp. $T_\Theta$ only (red), calibration data is used only in learning $T_X$, resp. $T_\Theta$.

While severe misspecifications in high dimensions may still require larger calibration sets, our results show that even small amounts of real data can substantially improve inference quality. This highlights the promise of our framework as a practical and scalable way to bring SBI closer to real-world scientific applications, and opens exciting opportunities for richer proposal architectures, domain adaptation techniques, and deployment on large-scale simulators where misspecification is inevitable.

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

## A   EXPERIMENTAL SETUP

We provide additional details on the training setup used across all experiments. For every task, we allocate a simulation budget of $N_{\text{sim}} = 5 \times 10^4$ samples and evaluate four calibration set sizes, $N_{\text{cal}} \in \{10, 50, 200, 1000\}$, which we denote by $N_1, N_2, N_3, N_4$.

During training, $20\%$ of each calibration set is reserved for validation and the remaining $80\%$ is used for training. We did not optimize over random seeds. All models were trained on a Nvidia RTX3060 GPU under 3 hours.

**Data preprocessing.** For tasks `Pendulum`, `WindTunnel`, and `LightTunnel` (uniform priors), we apply a logit transformation to map prior samples into $\mathbb{R}^p$. All datasets are standardized (z-scored) prior to training.

**Calibration sets.** For each $N_{\text{cal}}$, we generate 5 calibration sets by subsampling from a larger pool of calibration data. To reduce variance across runs, the sets are constructed in a nested fashion: denoting by $\mathcal{D}^i_{N_r}$ the $i$-th calibration set of size $N_r$, we enforce $\mathcal{D}^i_{N_r} \subset \mathcal{D}^i_{N_s}$ whenever $r < s$.

**Neural Posterior Estimation (NPE).** NPE is implemented with two components: a neural statistic estimator (NSE), $h_\omega(\mathbf{x})$, that encodes data into a low-dimensional representation, and a normalizing flow (NF) that maps a base distribution to the posterior. For task `Gaussian`, we use a standard neural spline flow (Durkan et al., 2019) and omit the NSE. For tasks `Pendulum`, `WindTunnel` and `LightTunnel`, we reuse the architectures and hyperparameters from Wehenkel et al. (2025).

**Flow Matching.** We use the architecture of Wildberger et al. (2023) as a backbone. For the $\boldsymbol{\theta}$-space flow $u_\Theta$, conditioning on $\mathbf{x}$ is implemented through a task-specific embedding network. For

the data-space flow $u_{\boldsymbol{X}}$, we employ the same architecture but with a separate embedding head for $(\mathbf{x}_t, t)$, equipped with positional encoding.

**Evaluation details.** The Wasserstein distance is computed using the `POT` package with default settings. For the C2ST, we train a classifier based on an MLP backbone, augmented with an embedding network for $\mathbf{y}$; the embedding architecture matches the one used for the normalizing flow. We apply 3-fold cross-validation and report the average validation accuracy across folds. By default, we balance the two classes - $C = 0$ (true samples) and $C = 1$ (generated samples)—but note that stratified K-fold can also be used to handle class imbalance.

## A.1 GAUSSIAN

The Gaussian task is defined as

$$p_{\boldsymbol{\Theta}} = \mathcal{N}(\mu_\theta, \Sigma_\theta), \quad p_{\boldsymbol{X}|\boldsymbol{\Theta}} = \mathcal{N}(A\boldsymbol{\theta} + b, \Sigma_x), \quad p_{\boldsymbol{Y}|\boldsymbol{\Theta}} = \mathcal{N}(C\boldsymbol{\theta} + d, \Sigma_y), \tag{10}$$

where

$\mu_\theta \in \mathbb{R}^3$, $\Sigma_\theta \in \mathbb{R}^{3 \times 3}$, $A \in \mathbb{R}^{10 \times 3}$, $b \in \mathbb{R}^{10}$, $\Sigma_x \in \mathbb{R}^{10 \times 10}$, $C \in \mathbb{R}^{10 \times 3}$, $d \in \mathbb{R}^{10}$, $\Sigma_y \in \mathbb{R}^{10 \times 10}$.

All parameters above are drawn randomly at the start of the experiment.

## A.2 PENDULUM

We follow the setup of Wehenkel et al. (2025, Appendix I.2). We sample $N = 200$ timesteps $t_i \sim \mathcal{U}[0, 10]$ and define the simulator

$$S : \begin{array}{rcl} \boldsymbol{\theta}, \epsilon & \longmapsto & [x_1, \ldots, x_N]^T \\ \text{with } x_i & = & A\cos(\omega_0 t_i + \varphi) + \epsilon_i, \quad \varphi \sim \mathcal{U}[0, 2\pi], \ \epsilon_i \sim \mathcal{N}(0, \sigma^2), \end{array} \tag{11}$$

where $\boldsymbol{\theta} = [A, \omega_0]$ are the parameters of interest. The high-fidelity data-generating process (DGP) includes damping:

$$\text{DGP} : \begin{array}{rcl} \boldsymbol{\theta}, \epsilon & \longmapsto & [y_1, \ldots, y_N]^T \\ \text{with } y_i & = & e^{-\alpha t_i} A\cos(\omega_0 t_i + \varphi) + \epsilon_i, \quad \alpha \sim \mathcal{U}[0, 1], \end{array} \tag{12}$$

where $\alpha$ encodes the friction coefficient. The prior is uniform: $p_{\boldsymbol{\Theta}} = \mathcal{U}[0, 3] \times \mathcal{U}[0.5, 10]$.

## A.3 WIND TUNNEL

We use the `load_out_0.5_osr_downwind_4` experiment from the `wt_intake_impulse_v1` dataset (Gamella et al., 2025). The data consist of 50-step time series measuring air pressure in the chamber after an impulse applied to the input fan. A hatch on the side controls an additional opening, which can be controlled with precision. The inference task is to predict its position $H \in [0, 45]$. We adopt model A2C3 from the `causalchamber` package as the simulator.

## A.4 LIGHT TUNNEL

We use the light tunnel experiment `uniform_ap_1.8_iso_500.0_ss_0.005` from the `lt_camera_v1` dataset (Gamella et al., 2025). A camera at the rear-end of an elongated chamber captures a light source emitted from the other end passing through two linear polarizers. We refer the reader to (Gamella et al., 2025) for more details about the mechanistic model of the tunnel. The inference task consists in inferring the color of the light source ($(R, G, B) \in [0, 255]^3$) as well as the Malus law (Collett, 2005) coefficient $\alpha \in [0, 1]$. The prior over these variables is uniform. The coefficient is a function of the polarizer angle $\alpha = \cos^2(\phi_1 - \phi_2)$, which are given in the dataset. The misspecification is introduced by omitting some physical aspects, which are detailed in (Gamella et al., 2025, Appendix D.IV.2.2).

# B ADDITIONAL PLOTS & METRICS

Additional metrics (MSE and MMD) are displayed for all previously compared methods, as well as for the following two additional other baselines from (Wehenkel et al., 2025),

- **Noisy Neural Posterior Estimation (NNPE)**: The amortized version of RNPE from (Ward et al., 2022), where the authors added Spike and Slab noise to the simulations and then trained a NPE model on these noisy data. This allegedly makes the method more robust to model misspecification by providing more conservative posterior estimation.

- **Joint Neural Posterior Estimation (J-NPE):** A posterior estimator was trained on both simulated and calibration data jointly, $\mathcal{D}_{\text{train}} = \{(\boldsymbol{\theta}_i, \boldsymbol{x}_i)\}_{i=1}^{N_{\text{sim}}} \bigcup \{(\boldsymbol{\theta}_j, \boldsymbol{y}_j)\}_{j=1}^{N_{\text{cal}}}$.

As the ground truth posterior distribution is available for the Gaussian task, posterior-level metrics are computed and reported in Figure 5, namely,

- **C2ST**: The regular C2ST as introduced in (Lopez-Paz & Oquab, 2017).

- $\boldsymbol{W_2 - \theta}$: The Wasserstein distance between parameter samples from the estimated posterior $\tilde{\boldsymbol{\theta}}_j \sim \tilde{p}(\boldsymbol{\theta}|\boldsymbol{y}_{\text{yobs}})$ for $1 \leq j \leq N_{\text{test}}$ and the true posterior $\boldsymbol{\theta}_i \sim p(\boldsymbol{\theta}|\boldsymbol{y}_{\text{yobs}})$ for $1 \leq i \leq N_{\text{test}}$ for a given observation $y_{\text{obs}}$.

$$W_2 = \min_{\gamma \in \mathbb{R}_+^{N_{\text{test}} \times N_{\text{test}}}} \left( \sum_{i,j} \gamma_{i,j} \|\boldsymbol{\theta}_i - \tilde{\boldsymbol{\theta}}_j\|_2^2 \right)^{\frac{1}{2}} \quad \text{such that } \gamma.\mathbf{1} = \frac{1}{N_{\text{test}}} \text{ and } \gamma^T.\mathbf{1} = \frac{1}{N_{\text{test}}}.$$

- $\boldsymbol{MMD - \theta}$: The Maximum Mean Discrepancy with a radial basis function (rbf) kernel between samples from the estimated posterior $\tilde{\boldsymbol{\theta}}_j \sim \tilde{p}(\boldsymbol{\theta}|\boldsymbol{y}_{\text{yobs}})$ for $1 \leq j \leq N_{\text{test}}$ and the true posterior $\boldsymbol{\theta}_i \sim p(\boldsymbol{\theta}|\boldsymbol{y}_{\text{yobs}})$ for $1 \leq i \leq N_{\text{test}}$ for a given observation $y_{\text{obs}}$.

For all other tasks, metrics are computed between joint distribution samples and reported in Figure 6.

Additionally, KDE plots for tasks `WindTunnel` and `LightTunnel` are shown, respectively, in Figures 7 and 8. RoPE was omitted in Figure 8 as it hindered clear visualization of the other posterior distributions due to a too flat posterior estimate.

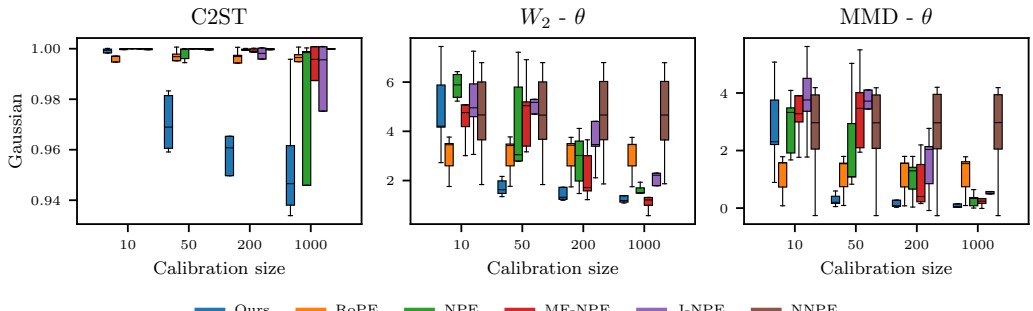

Figure 5: Parameter sample metrics ($\downarrow$ is better). C2ST, Wasserstein distance and Maximum Mean Discrepancy (MMD) for the Gaussian task. For a given observation $\boldsymbol{y}_{\text{obs}}$, we sampled $N = 5000$ parameters $\{\boldsymbol{\theta}_i\}_1^N$ for each method. Each boxplot show the distribution of metrics value over 3 different observations $\boldsymbol{y}_{\text{obs}}$ chosen at random.

## C  TRAINING AND INFERENCE TIME

We report the training (Table 1) and inference (Table 2) times for our method and the baselines compared in Figure 2. Table 3 provides a detailed breakdown of the training time of our method across its main components. Although our method is more expensive to train than the baselines, it remains practical, requiring only 5–6 minutes per task. As shown in Table 3, most of the training cost arises from the flow-sampling step (ODE integration), which is not optimized in our current implementation. We use a basic explicit-Euler solver, whereas more advanced integrators (e.g., midpoint methods, higher-order adaptive solvers, or JAX-based JIT compilation) could significantly reduce the number of integration steps and thus speed up training. Other methods such as distillation

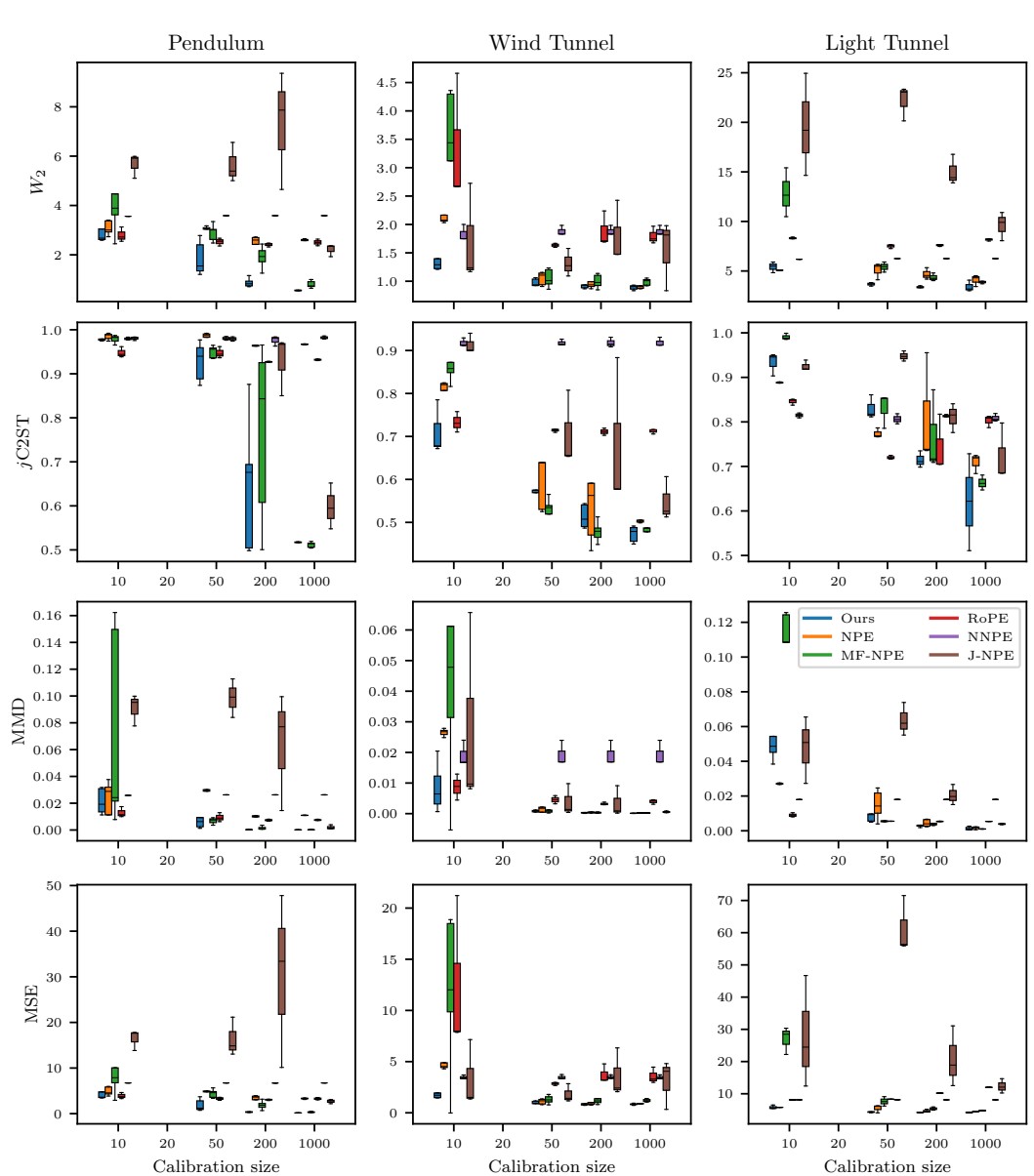

Figure 6: Joint sample metrics (↓ is better). Wasserstein distance (top row), $jC2ST$ (second row) MMD (third row) and MSE (bottom row) with respect to an increasing calibration set size $N_{cal} \in \{10, 50, 200, 1000\}$. Each boxplot shows the distribution of metric values across five independent runs, each using a different randomly chosen calibration set.

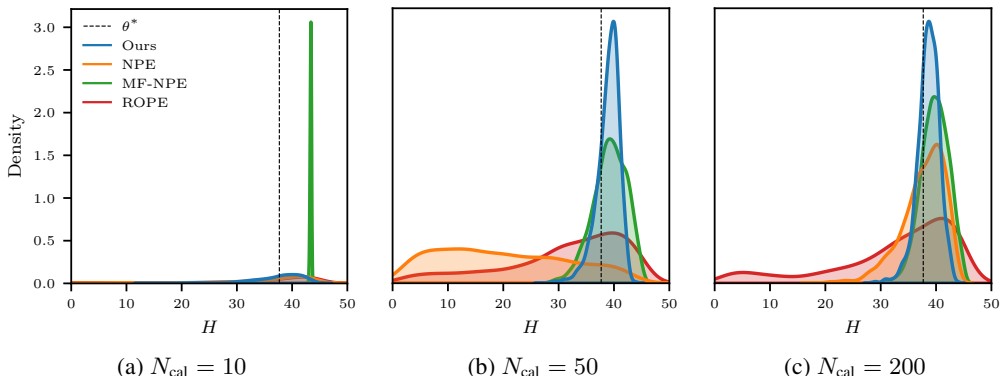

(a) $N_{\text{cal}} = 10$          (b) $N_{\text{cal}} = 50$          (c) $N_{\text{cal}} = 200$

Figure 7: Kernel density estimates of the learned posteriors for task `WindTunnel`. For a given $\boldsymbol{y}^* \in \mathcal{D}_{\text{test}}$, we draw $\{\hat{\boldsymbol{\theta}}_i\}_{1 \leq i \leq 2000}$, for each method and 3 calibration sizes $N_{\text{cal}} \in \{10, 50, 200\}$. The dotted black line indicates the true parameter $\boldsymbol{\theta}^*$ that generated $\boldsymbol{y}^*$.

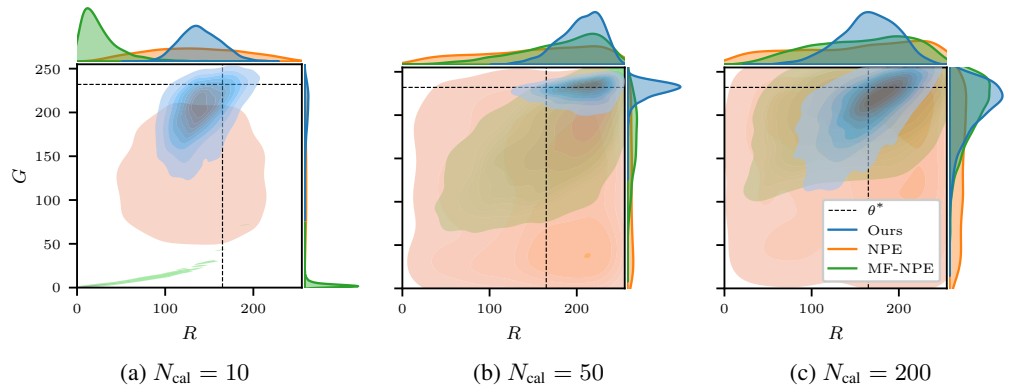

(a) $N_{\text{cal}} = 10$          (b) $N_{\text{cal}} = 50$          (c) $N_{\text{cal}} = 200$

Figure 8: Kernel density estimates of joint and marginal samples for task `LightTunnel`. We report the posterior densities for the first two coordinates of the parameter $[\theta_1, \theta_2] = (R, G)$. For a given $\boldsymbol{y}^* \in \mathcal{D}_{\text{test}}$, we draw $\{\tilde{\boldsymbol{\theta}}_i\}_{1 \leq i \leq 2000}$, for each method and 3 calibration sizes $N_{\text{cal}} \in \{10, 50, 200\}$. Dotted black lines indicate the true parameter $\boldsymbol{\theta}^*$ that generated $\boldsymbol{y}^*$.

(Dao et al., 2025) or straight flow trajectories (Kornilov et al., 2024) can help us reduce the training time even further. At inference time, our method is faster than RoPE for all tasks and calibration sizes.

Table 1: Training time (in seconds) for each method and calibration set size $c$. For the Gaussian task, no training time is reported for RoPE as no embedding network is used for NPE.

| Method | Gaussian | Light Tunnel | Pendulum | Wind Tunnel |
|---|---|---|---|---|
| **c = 10** | | | | |
| NPE | 5.51 | 0.79 | 0.63 | 0.58 |
| Ours | 28.35 | 37.12 | 52.82 | 94.73 |
| MF-NPE | 3.48 | 0.55 | 0.72 | 1.53 |
| RoPE | – | 5.40 | 7.81 | 7.10 |
| **c = 50** | | | | |
| NPE | 6.07 | 0.71 | 0.26 | 0.14 |
| Ours | 63.29 | 70.97 | 80.74 | 126.74 |
| MF-NPE | 6.33 | 1.76 | 0.76 | 1.92 |
| RoPE | – | 6.53 | 7.95 | 7.67 |
| **c = 200** | | | | |
| NPE | 13.11 | 4.11 | 2.21 | 0.19 |
| Ours | 70.24 | 136.18 | 163.91 | 127.65 |
| MF-NPE | 10.35 | 4.38 | 2.86 | 3.02 |
| RoPE | – | 15.93 | 11.87 | 11.73 |
| **c = 1000** | | | | |
| NPE | 37.32 | 15.10 | 12.32 | 5.94 |
| Ours | 141.96 | 306.55 | 333.70 | 235.78 |
| MF-NPE | 22.33 | 26.76 | 8.45 | 10.99 |
| RoPE | – | 61.57 | 38.09 | 38.47 |

Table 2: Inference time (in seconds) for each method and calibration set size $c$. For each method and task, we evaluated a test set of $N_{\text{test}} = 5000$ real-world observations $\{y_i\}_{1 \leq i \leq N_{\text{test}}}$. The reported times correspond to the total duration to generate all 5000 corresponding parameters.

| Method | Gaussian | Light Tunnel | Pendulum | Wind Tunnel |
|--------|----------|--------------|----------|-------------|
| **c = 10** | | | | |
| NPE | 0.09 | 0.67 | 0.12 | 0.02 |
| Ours | 0.64 | 10.95 | 2.20 | 0.38 |
| MF-NPE | 0.01 | 0.69 | 0.06 | 0.01 |
| RoPE | 110.31 | 222.51 | 446.48 | 54.80 |
| **c = 50** | | | | |
| NPE | 0.01 | 0.63 | 0.07 | 0.01 |
| Ours | 0.73 | 10.81 | 2.22 | 0.41 |
| MF-NPE | 0.01 | 0.70 | 0.06 | 0.01 |
| RoPE | 106.94 | 218.01 | 93.07 | 54.32 |
| **c = 200** | | | | |
| NPE | 0.01 | 0.64 | 0.07 | 0.01 |
| Ours | 0.55 | 10.83 | 2.21 | 0.43 |
| MF-NPE | 0.01 | 0.58 | 0.08 | 0.01 |
| RoPE | 101.60 | 224.80 | 87.91 | 56.14 |
| **c = 1000** | | | | |
| NPE | 0.01 | 0.64 | 0.06 | 0.01 |
| Ours | 0.40 | 11.03 | 2.20 | 0.43 |
| MF-NPE | 0.01 | 0.65 | 0.08 | 0.01 |
| RoPE | 100.11 | 226.29 | 88.23 | 59.13 |

Table 3: Training time (in seconds) for our method (FMCPE) broken down into 3 components: (1) Gradient Step: time to perform gradient updates on the calibration set; (2) Sample NPE: time to sample from the base NPE model; (3) Sample x Flow: time to sample from the learned flow in the observation space.

| Task | Component | $c = 10$ | $c = 50$ | $c = 200$ | $c = 1000$ |
|------|-----------|----------|----------|-----------|------------|
| Gaussian | Gradient Step | 0.18 | 0.68 | 0.84 | 1.67 |
| | Sample NPE | 8.96 | 19.84 | 21.70 | 43.54 |
| | Sample $x$ Flow | 19.20 | 42.76 | 47.70 | 96.75 |
| Light Tunnel | Gradient Step | 0.68 | 1.43 | 2.36 | 2.81 |
| | Sample NPE | 9.82 | 18.24 | 33.68 | 58.25 |
| | Sample $x$ Flow | 26.62 | 51.29 | 100.14 | 245.49 |
| Pendulum | Gradient Step | 0.94 | 1.86 | 3.74 | 5.97 |
| | Sample NPE | 11.04 | 16.75 | 34.45 | 70.04 |
| | Sample $x$ Flow | 40.83 | 62.12 | 125.72 | 257.69 |
| Wind Tunnel | Gradient Step | 1.10 | 1.68 | 1.84 | 2.90 |
| | Sample NPE | 21.03 | 27.93 | 27.91 | 52.04 |
| | Sample $x$ Flow | 72.60 | 97.13 | 97.90 | 180.83 |

