# OpenReview forum: "Flow Matching for Robust Simulation-Based Inference under Model Misspecification"
_ICLR.cc/2026/Conference — Submitted to ICLR 2026_

### Official Review · Reviewer_TKbH · 2025-10-26

**Soundness:** 2
**Presentation:** 3
**Contribution:** 2
**Rating:** 4
**Confidence:** 3

**Summary:**

This paper aims to solve the model misspecification problem under simulation-based inference (SBI) with flow matching. It proposes a two-stage flow-matching scheme: (i) an observation-space flow $T_X$ that transports Gaussian perturbations around a real observation y toward a proxy distribution q(x∣y), and (ii) a parameter-space flow $T_\\theta$ that maps samples from a source distribution built from an amortized posterior into an approximation of the true posterior. Across several toy and real-world tasks, the method appears robust in low-calibration-data regimes and improves downstream metrics vs. amortized baselines.

**Strengths:**

* The proposed method is quite interesting. The construction of $\\pi(\\theta∣y)$ via $T_X$ reduces the transport gap and stabilizes training in small-calibration settings.
* To my understanding, it works as a post-hoc calibration layer on top of any amortized posterior estimator and does not rely on restrictive conditional-independence assumptions like some of the previous methods.
* The empirical result looks solid, which includes multiple tasks, and shows consistent gains as calibration size grows.
* The presentation is clear, also the source code is provided.

**Weaknesses:**

* Lack of robust SBI baselines: Although the paper argues RoPE is “not directly comparable,” I believe it is still a meaningful baseline because the setups are closely related: both methods correct amortized posteriors under simulator–real mismatch using a small calibration set and a learned transport/alignment step. The key difference is protocol (RoPE is transductive—using the test set as a whole—whereas your method is inductive—per-sample), but that does not preclude comparison; it simply requires transparent labeling. A practical compromise is to report RoPE under its native transductive protocol (clearly marked as such) alongside your inductive results. This lets readers gauge the attainable performance when the full test set is accessible, while still appreciating your method’s per-sample advantages. In addition, adding other previous robust SBI baselines like what RoPE did would further strengthen the paper—e.g., NNPE, NPE-RS, or J-NPE that jointly trains on simulated and calibration pairs.
* Missing ablations: The paper motivates each module but does not quantify its contribution. It would be great if the authors could add some ablation studies (some may not make sense)
  * X flow only without theta flow, then you could feed $\\tilde{x}$ into a standard NPE.
  * theta flow only. You could feed the pre-trained NPE directly with y, obtain $\\hat{\\theta}$, and use your theta flow to map them to the target.
  * Gaussian start. sample theta_0 from a Gaussian, learn only theta flow with few calibration pairs.
  * Sequential vs joint training. Is joint training necessary? What if we train them separately?
* It would be great to add a few more metrics, such as MMD, and also coverage/ECP for posterior calibration.

**Questions:**

* jC2ST scores below 0.5 in wind tunnel example is a bit unusual to me given N_test = 5000. Could you give more explanation about it?
* A Gaussian $N(y, \\sigma^2I)$ is used to sample $x_0$, I think it needs some clearer justification, and how sensitive are results to $\\sigma$?

---

> ### Author Response · Authors · 2025-11-26
>
> We thank the reviewer for his/her assessment of our work and for finding that our “proposed method is quite interesting” and “the presentation is clear”. We are also grateful for his/her appreciation of our empirical results (“empirical results look solid”). We address below the concerns raised by the reviewer.
>
> ## Lack of baselines
>
> We added a comparison to RoPE as well as other baselines in the results; they are available at this link (https://figshare.com/s/f5938eb4836861483dd1) and will be added in a new version of our manuscript to be uploaded soon. Please see also our response to reviewer q8cA for more details and discussion.
>
> ## Ablation study
>
> We have run an ablation study considering:
> - Sequential Training : Instead of the joint minimization process, we first train the map $T_X$ until convergence (same flow matching loss) and we then learn the map $T_{\theta}$ using the learned $T_X$ to compute the proposal.
> - NPE + $T_X$: We only train the map $T_X$ on calibration data and feed the obtained $\tilde{x}$ into the NPE directly. Note that in RNPE (Ward et al. 2022), the authors define their corrected posterior in a similar manner, but instead of a learned mapping $T_X$ to obtain $\tilde{x} = T_X(y)$, they use MCMC to sample from a fixed $p(x|y)$, Spike and Slab type of noise.
> - $T_{\theta}$ + NPE: We only train the map $T_{\theta}$ using the NPE trained on simulation data evaluated on the real data $y$. The source distribution writes $\pi(\theta | y) = \hat{p}_{\Theta|X}(\theta | X = y)$.
>
> We have included a figure in the Appendix with results on three examples: pendulum, wind tunnel, and light tunnel. We have not included the Gaussian example because all curves seemed to overlap and did not convey much insight. See this link (https://figshare.com/s/733e35cddaf874331db3) for a preview.
>
> ## More metrics
>
> We have included results in terms of MMD in the Appendix of the new version of our manuscript.
>
> ## jC2ST smaller than 0.5
>
> We agree that a jC2ST smaller than 0.5 may sound odd and we have included a short sentence in our manuscript on this topic. Indeed, a dataset with $N_{test}=5000$ samples can be quite small to train a classifier on the joint distribution $(\theta,y)$, which can lead to intrinsic problems in the jC2ST metric calculation; this is one of the reasons why we have included other metrics to support our findings.
>
> ## Gaussian source distribution for the $T_X$ flow
>
> In the misspecification context, simulations and observations are assumed to have relatively high mutual information. Thus in a wide range of applications and considering the low amount of training data, starting from $y$ facilitates the learning process. This choice only works when $x$ and $y$ have the same dimension. We did not notice a significant change in performance when varying sigma in a reasonable range ($[0.01,1]$) but we suspect that further analysis taking into account some sort of signal-to-noise ratio between x and y could be useful in the future.

---

### Official Review · Reviewer_92qm · 2025-10-30

**Soundness:** 3
**Presentation:** 3
**Contribution:** 2
**Rating:** 4
**Confidence:** 3

**Summary:**

The paper studies multifidelity simulator-based inference, where the goal is to infer the posterior distribution of model parameters $\theta$ given the low-fidelity data $(x,\theta) \sim p(x,\theta)$ and the limited high-fidelity data $(y,\theta) \sim p(y,\theta)$ (termed as calibration data). The paper proposes training two flows (i) a flow from the base distribution to $q(x|y)$ using the calibration data, and (ii) a flow from the misspecified posterior of $\theta$ to the correctly specified posterios of $\theta$, using joint flow-matching. The method shows good performance compared to some exististing baselines  in synthetic and real-world tasks,  in terms of quantative and qualitive metrics.

**Strengths:**

The paper is well-written, easy to follow, and the proposed method is cleary articuled in Section 3, where I could not find any tehcnical flaws or ad-hoc justifications. While there are some limitations in the experimental section (see discussion in Weaknesses), the paper demontstrate good and relatively robust emprical performance in synthetic and real-world tasks. When evaluated in the context of multifidelity SBI, the proposed method constitutes a contribution in terms of novel ideas and empirical performance.

It is great that the experimental section contains visualizations of the learned posteriors for both the baselines and the proposed method. This makes it easier to qualitatively judge performance and shows that the proposed method produces more accurate and robust posteriors than the baselines.

**Weaknesses:**

The main weakness is the insufficiently clear problem formulation, which results in a somewhat exaggerated scope. I am quite convinced that the paper does not tackle SBI under model misspecification but rather multifidelity SBI, where in the latter there is some information about the true model as we have access to the samples $(y,\theta)$. I try to elaborate this weaknesses below:

Lines 78-82: “…high-fidelity data—accurate representations of the phenomenon obtained either from costly high-quality simulations or from ground-truth observations.”
Isn’t the application scope quite limited in the setting where the ground-truth observations are samples from the joint $p(\theta,y)$ rather than from the marginal $p(y)$? Typically, $\theta$ represent latent variables that only exists within the model. For that reason, the considered problem in the paper is not a classical SBI problem but rather a multi-fidelity SBI problem, the term that already appears in the title of (Krouglova et al., 2025). Of course, this same critique applies to (Wehenkel et al., 2025), but I think it is important to be explicit about this to avoid misunderstanding that the proposed method is a general SBI method (as e.g. the title currently hints). This fact is also reflected in the experiments where only real-world tasks are from (Gamella et al., 2025).

Given this, I suggest:
1.	The title should replace the term “SIMULATION-BASED INFERENCE” to “MULTIFIDELITY SIMULATION-BASED INFERENCE” to honestly capture the scope of the paper.
2.	The accommodiate the first sentences in Section 3: “We consider the general problem of sampling from a posterior distribution…” to something where the paper early states that ‘”…we have access to samples $(y,\theta)$”.

Next, I discuss some minor weaknesses.

Evaluation metrics (Section 4.2)

This section lacks justification and discussion for the chosen evaluation metrics. For example, if the task is to infer the posterior of $\theta$, why the paper does not consider the distance (say Wasserstein) between the inferred  posterior and the ground-truth $p(\theta)$. This should be feasible at least in the task “Gaussian”, where the ground-truth $p(\theta)$ is known. Further, why the paper considers MSE of $\theta$ samples, while we are dealing with distributions, isn’t some metric in the space of probabilitdy distributions (Wasserstein etc.) more appropriate?

Baselines

“….we do not include RoPE (Wehenkel et al., 2025) because it is a method that requires access to the full test set at inference time and is not directly comparable to our approach, as explained in Section 2.” The conditional independence assumption imposed by (Wehenkel et al., 2025) it maybe a limitation of their work but should not be a reason to exclude RoPE from the experiments. Further, can you elaborate why requiring access to the full test set at the inference time is a critical limitation? Concernign the sequential setup, I think that all the experiments in the paper are non-sequential, right?

Experimental results

In real-world tasks (Wind tunnel, Light tunnel), and when considering the metric that focuses on the posterior samples of $\theta$ (i.e. MSE), the all the methods looks achieving comparable performance when “Calibration size” is not very small (Figure 4). It would be good to discuss this in the main paper

**Questions:**

Can you list some real-world problems that fits to the problem considered in the paper? I tried to quickly check (Wehenkel et al, 2025), but so far could only confirm that it matches with (Gamella et al., 2025).

---

> ### Author Response · Authors · 2025-11-26
>
> We thank the reviewer for his/her assessment of our work. We were happy to learn that our “paper is well-written” and “easy to follow”, as well as that there were “no technical flaws” to be found. It is also pleasant to read that our “proposed method constitutes a contribution in terms of novel ideas and empirical performance.” Here below we address the main concerns and questions raised by the reviewer.
>
> ## Clarification between multi-fidelity vs misspecification
>
> As we explain in our response to reviewer `q8cA`, we regard misspecification as a property of the simulator, not as something tied to whether calibration data are ground truth or simulated, nor to their availability. Please refer to that response for further detail.
>
> Thanks for suggesting a new title to clarify this point. We would rather propose to update the manuscript title to “Flow-Matching **CALIBRATION** for Robust SBI under Model Misspecification”, which explicitly reflects that the approach can equally incorporate high-fidelity simulated or  ground-truth calibration data when available. We have also added clarifying explanations  in the revised manuscript (to be uploaded soon) addressing the reviewer’s concerns more directly.
>
> ## Choice of evaluation metrics
>
> We added the Wasserstein distance between the true and estimated posteriors as well as the C2ST score for the Gaussian task. See the new figure at this link (https://figshare.com/s/f5938eb4836861483dd1). We have also added MMD results in the Appendix.
>
> ## Comparison with RoPE
>
> We agree with the reviewer that our analysis is more complete when including RoPE as a baseline. We have now added results with this method, as well as with the other baselines proposed in Wehenkel et al. (2025). Further details are provided in our response to reviewer `q8cA`. A figure with the new results is available at this link (https://figshare.com/s/f5938eb4836861483dd1), where we see that although RoPE requires having access to the full test set at inference time, on top of fresh simulations ${x_i}$ with ${1 \leq i \leq N_s}$ to compute an OT matrix, its performance is not better than our proposal.
>
> Regarding the comment about a sequential versus non-sequential setup, we are unfortunately not certain we have interpreted the reviewer’s point correctly. Could the reviewer please clarify what specific distinction he/she had in mind? This would help us address the concern accurately in the next revision.
>
> ## Comparable performance of all methods with high $N_{cal}$
>
> Whenever the calibration dataset starts to inflate too much, one can expect that a naive NPE estimation based directly on the pairs $(\theta, y)$ would have enough data to have acceptable performance, as seen in the figures. Indeed, one of the main “sanity checks” that we used when conceiving FMCPE was to make sure that it was always better than naive-NPE for small calibration datasets.
>
> ## Real world problems application
> We agree that ground-truth parameters are not available in every application. However, having a calibration dataset with known parameters is not an uncommon situation. In many real domains, the parameters are simply the controlled inputs of the experiment—for example in industrial process control, manufacturing, robotics, microfluidics, electronics, or controlled neuroscience stimuli. In these settings, $\theta$ is known because it is set and logged by the operator, and expensive calibration runs are already part of standard practice for tuning or validating simulators.
>
> Our method only requires such a moderate calibration set, not ground truth for all observations, and therefore remains applicable to a broad class of practical scenarios where controlled experiments or reference runs are routinely available.

---

### Official Review · Reviewer_iYZs · 2025-10-30

**Soundness:** 3
**Presentation:** 3
**Contribution:** 2
**Rating:** 4
**Confidence:** 4

**Summary:**

This paper proposes a method to address model misspecification in SBI via flow matching trained on a calibration dataset with known ground truths. I am short on time due to the semester start. Apologies if my reviews are a bit short. I am happy to engage in reviewer discussion should be concerns not be clear. And I am happy to consider increasing my score should the authors provide convincing responses to my concerns.

**Strengths:**

- The paper addresses the important problem of model misspecification in SBI/ABI.
- The method appears theoretical sound and overall sensible to me.
- I think the paper is well written. But this assessment may be different for readers less familiar with the field of SBI/ABI.

**Weaknesses:**

- Assuming the existance of a calibration dataset with known ground truths is a strong assumption. I know several papers make this assumption. But still, for many application areas we just never have the ground truth parameters available ruling out the proposed method for their analyses.
- The tested benchmarks are all very low dimensional in parameter space. How does the size of the required calibration dataset scale with the parameter dimensionaliry of the problem? I would have expected this scalability to be analyzed at the very least in toy examples (Gaussian).
- Some relevant related work was not cited or considered as benchmarks (at least https://arxiv.org/abs/2501.13483 and https://arxiv.org/abs/2502.04949). Perhaps the references therein provide even more relevant research?
- A comparison of both training and inference speeds between the approaches is lacking to my reading. At least I haven't found it. Providing information in that regard would be important too I think.

**Questions:**

- Just to double check: The approach is fully amortized right? In the sense that, at inference time, only forward passes through the networks are required without any (re-)training?
- What is the target of inference? The true posterior under the misspecified model or the true posterior under the correctly specified model? See also https://arxiv.org/abs/2502.04949 for a taxonomy of interence targets in the face of misspecification.
- How strongly is the idea of the method rooted in flow matching? I.e., how easy would it be to replace the flow matching networks with standard diffusion models or even coupling flows?

---

> ### Author Response · Authors · 2025-11-26
>
> We thank the reviewer for his/her assessment of our work. We appreciate the reviewer acknowledging the importance of SBI misspecification and soundness of the method that we propose. We are also delighted to learn that, according to the reviewer, our paper is well written. We address the concerns and questions below.
>
> > Assuming a calibration dataset with known ground truths is a strong assumption. In many application areas, we just never have the ground truth parameters available ruling out the proposed method for their analyses.
>
> We agree that ground-truth parameters are not available in every application. However, having a calibration dataset with known parameters is not an uncommon situation. In many real-world domains, the parameters are simply the **controlled inputs of the experiment**—for example in industrial process control, manufacturing, robotics, microfluidics, electronics, or controlled neuroscience stimuli. In these settings, $\theta$ is known because it is set and logged by the operator, and expensive calibration runs are already part of standard practice for tuning or validating simulators.
>
> Our method only requires such a moderate calibration set, not ground truth for all observations, and therefore remains applicable to a broad class of practical scenarios where controlled experiments or reference runs are routinely available.
>
> > The tested benchmarks are all very low dimensional in parameter space. How does the size of the required calibration dataset scale with the parameter dimensionality of the problem?
>
> This is clearly an interesting question that we have not yet considered. Most examples in the SBI literature are of small to moderate parameter dimensions. However, we can expect the requested calibration size to increase with the parameter dimension, but a proper theoretical and experimental study would require further work and was out of the scope of this paper.
>
> > Some relevant related work was not cited or considered as benchmarks
>
> We thank the reviewer for these two very interesting references related to Unsupervised Domain Adaptation (UDA). The addition of a Bayesian self-consistency loss in [Mishra et al 2025] is a great and inspiring idea. However, in practice, it seems to mainly address out-of-distribution misspecification, e.g. when parameters are not simulated in the right domain or from a wrong prior.
>
> The Bayesian self-consistency loss requires an analytical expression of the likelihood or a good approximation of it, which thus limits the kind of misspecification that can be handled and the class of generative models underlying the approximations. However, the ability of making use of ground truth observations is a nice feature that could be interesting to couple with procedures like ours. This would require further investigations. We will add a discussion on these references.
>
> > Comparison of both training and inference speeds between the approaches is lacking to my reading
>
> We added a detailed report in the Appendix of every method training and inference speed with a breakdown of each component training time for our method (Simulator call, ODE step and gradient step). See Tables at this link (https://figshare.com/s/0edfe75b333f8500a40b) for a preview.
>
> > Q : Is the approach fully amortized?
>
> Yes, our approach is indeed fully amortized. We do not need any additional training step to sample from an unseen observation $y$. We apologize if this important feature of our methodology was not sufficiently clear and have included more explicit mentions of it in the revised text.
>
> > Q : What is the target of inference?
>
> The target of our inference pipeline is the true posterior under the well-specified model, which corresponds to target 1 of (https://arxiv.org/abs/2502.04949). We will clarify this in the new version of our manuscript.
>
> > Q : How strongly is the idea of the method rooted in flow matching?
>
> Our method is not inherently tied to the Flow Matching framework; rather, it works as a convenient tool to model conditional distributions. There are, however, a few reasons to use Flow Matching over other methods such as diffusion and discrete normalizing flows. On the first hand, our approach of using a non-gaussian source distribution to learn the parameter map $T_{\Theta}$ doesn’t fit naturally in the diffusion framework. Normalizing flows are more flexible to using a non-gaussian source distribution, but they still require evaluating the log-probability of said source distribution, which flow matching doesn’t require.

---

> > ### Comment · Reviewer_iYZs · 2025-11-26
> >
> > Thank you for your detailed answers. I decided to keep my score for now but will observe also the responses of other reviewers and may change my mind before the end of the discussion period.

---

### Official Review · Reviewer_q8cA · 2025-10-31

**Soundness:** 3
**Presentation:** 1
**Contribution:** 2
**Rating:** 2
**Confidence:** 4

**Summary:**

The paper proposes **FMCPE**, a two-stage flow-matching correction for SBI under misspecification: train an NPE on simulations, then learn two conditional flows—one in data space $T_X$​ and one in parameter space $T_\Theta$—to transport toward a “corrected” posterior using scarce calibration pairs. The authors consider a *joint training* objective of corrected posterior and source distribution. Practical (Algorithm 2) each training tuple requires a **simulator call** ​ and an **ODE solve** to estimate $T_x$ jointly with a correction map $T_\Theta$ the adjust a pretrained NPE (on low-fidelity or misspecified data) (Algs. 1–2). The paper then empirically validates the proposed approach on several misspecification-tasks from previous work (Wehenkel et.al. 2025) and demonstate empirical advantages against a naive NPE baseline (only trained on calibration set) and a MF-NPE (pretrained on simulations, then fine tune on calibration).

**Strengths:**

- **Novelty**: To my knowledge this learning-based approach is novel and address limitations of previous approaches  which relied on pre-specified form of misspecification or conditional independence assumptions.

**Weaknesses:**

1. **Conceptual positioning (misspecification vs. multifidelity/multilevel):**
    The method and experiments read sometimesmore like a **multifidelity/multilevel** correction pipeline (low-fidelity simulator + scarce high-fidelity pairs + learned transport) than a principled treatment of misspecification. Additionally the comparisons target **NPE** and **MFNPE**; there is **no direct baseline targeted at misspecification** (e.g., methods that explicitly correct simulator bias), and RoPE is excluded from evaluation. This creates an **identity gap** between the stated aim (misspecification) and what is empirically validated (multifidelity fine-tuning). The authors consider the same tasks as in Wehenkel et al. 2025, but claim that its not "comparable" because it requires the full test set at inference time. While I agree that this is a limitation that the proposed method addresses, this **does not** prevent a comparison!

2. **Computational cost & performance:**
    Every training tuple triggers **both** a **simulation** **and** an **ODE integration**. This pipeline seems **extremely costly** compared to single-flow or single-stage amortized approaches and scales poorly with calibration size. Analysis or discussion of this is completely missing in the manuscript. Furthermore it should come with strong performance gains, specifically it **must** be better than methods just using e.g. the calibration dataset (similar to what the authors demonstrate with NPE). Recent work [1] has show that Tabular Foundation models can do SBI with improved robustness and simulation-efficiency. A good way to demonstrate the efficacy would be to include a comparison against such methods that e.g. only "train" (evaluate) on the calibration datasets (or maybe mixed with cheap simulations similar to MF-NPE).

[1] Vetter, Julius, et al. "Effortless, Simulation-Efficient Bayesian Inference using Tabular Foundation Models." _arXiv preprint arXiv:2504.17660_ (2025).

3. **Weak metrics and limited empirical coverage:**
        The paper evaluates via **joint** discrepancy metrics (jC2ST, Wasserstein on the joint), rather than **posterior-level** diagnostics that directly assess the **misspecified posterior** when that posterior is tractable or obtainable via MCMC (which is true for at least the Gaussian task). This makes the evidence **indirect** and weaker than necessary. For example this are effectively very similar to *global coverage* metrics which are only *necessary* but **not** sufficient properties that can be fooled by conservative posterior approximations (i.e. any convex combination of the posterior and prior). The empirical evidence in general is additionally limited to 4 tasks. While the authors claim that the method does not require the conditional independence assumption required for rope (and I agree conceptually), its unclear if the presented tasks demonstrate this.

**Questions:**

- **Multi-fidelity vs. misspecification.** These are related but to a some degree also quite different concepts, and the distinction should be communicated clearly by explicitly defining both terms in the text. In misspecification (e.g., sim-to-real gaps), calibration datasets are typically rare or impossible to obtain; by contrast, in multi-fidelity/multilevel settings they exist by definition. Moreover, simulators are generally misspecified to some degree and need not be “cheap” to run, whereas in multi-fidelity setups usually include lower-cost simulators that would justify the additional simulation burden of the proposed approach (but this is not necessarily given in misspecified settings). This should be clearly communicated in the manuscript.
- Can you add **model-misspecification baselines** i.e. ROPE or others (not just multi-fidelity ones) and report **posterior-level** metrics where the ground-truth misspecified posterior is tractable?
- Another recent multi-fidelity approach needs to be discussed [1].
- What is the **wall-clock/NFE** budget per calibration size, broken down into simulator calls vs. **odeint** solves​ and flow training steps?


[1] Hikida, Yuga, et al. "Multilevel neural simulation-based inference." _arXiv preprint arXiv:2506.06087_ (2025).

---

> ### Author Response · Authors · 2025-11-26
>
> We thank the reviewer for his/her thoughtful and thorough evaluation of our paper, as well as for the valuable suggestions he/she provided. Below, we address the main identified weaknesses and respond to all questions.
>
> ## Multi-fidelity vs misspecification:
>
> We fully understand the remark and are a bit puzzled that our presentation was perceived as problematic in that regard, as also pointed out by reviewers `iYZs` and `92qm`.
>
> Our general feeling is that the two terms, **multifidelity and misspecification, are not comparable concepts** and thus do not have to be exclusive or distinguished from one another, and thus may not lead to two clearly distinct categories of methods.
>
> Indeed, *misspecification* refers to the underlying problem, whose most critical manifestation in applications is bias in the posterior and the resulting inferences. *Multifidelity*, by contrast, denotes a potential tool or approach to address or mitigate this problem. In that sense, the two concepts are not interchangeable: one describes the issue, the other a possible solution. They can therefore be used together—or mentioned separately—without risk of confusion.
>
> That said, according to the reviewer’s own words, *misspecification* refers to methods that explicitly correct simulator bias. Such correction is exactly what our paper targets. And this is what characterises our attempt independently on the availability of ground truth pairs or not. In our point of view, misspecification is a property of the simulator, not a feature characterizing the nature (ground truth or simulated) or availability (missing or not missing) of calibration data.
>
> **So why not refer more explicitly to our method as a multi-fidelity one?**
>
> We decided to rebrand FMCPE as a **calibration** approach for the following reasons:
>
> 1) **Our setting involves one simulator + a fixed calibration dataset, not multiple simulators.** Multifidelity methods require access to several simulators (e.g., low- and high-resolution). In our case, the calibration data are not another simulator: they are measurements from the real system or from a more accurate—but not accessible—model. They cannot be generated on the fly, and no high-fidelity simulator is assumed to be available on demand. In addition, our calibration set does not have to be seed-matched to the simulations as required in MLMC-based procedures, e.g. Hikida et al. (2025).
>
> 2) **Multifidelity focuses on simulation strategies, or requires the existence of a hierarchy of nested simulators**, whereas our setup is static and does not require any link between simulated and calibration data. As highlighted in (Peherstorfer et al. 2016, Prescott et al. 2024) multifidelity methods typically decide which fidelity level to run during simulation, optimising accuracy-cost trade-offs under a fixed budget. They also often require simulators to be nested and of increasing precision, e.g. to apply MLMC strategies such as in Hikida et al. (2025). Our setting differs: the calibration set is given and fixed and does not have to be simulated from the same random seeds or from related models. That said, some workflows treat real data as “highest-fidelity” and the simulator as lower-fidelity. In that extended sense, one could describe our setup as multifidelity calibration, but technically it is a misspecified SBI problem with access to calibration data, and the notions are not contradictory.
>
> 3) **Misspecification is the core problem we target; calibration is the cure we employ.** As stated in the abstract and introduction (line 47), misspecification arises when simulators only approximate reality, producing simulated data that do not match observations and thus lead to biased posteriors. Our focus is on treating these symptoms. Multifidelity is one possible remedy, and we adopt it by interpreting the calibration data as “high-fidelity” observations (lines 80, 86, 135, 155).
>
> 4) **We state that calibration data may as well come from real observations or high-fidelity simulations.** This appears in several places (lines 80, 86, 135, 155). However, we acknowledge that we accidentally referred to calibration as “real” data in two key locations (abstract and conclusion, line 469). We will correct this wording.
>
> Considering all this and all reviewers comments, we are willing to improve our presentation, as suggested.
>
> [Prescott et al. 2024] Efficient multifidelity likelihood-free Bayesian inference with adaptive computational resource allocation. Journal of Computational Physics, 2024.
>
> [Peherstorfer et al. 2016] Optimal model management for multifidelity Monte Carlo estimation. SIAM J. Sci. Comput, 2016.

---

> ### Author Response · Authors · 2025-11-26
>
> ## Comparison with RoPE
>
> We agree that although RoPE requires the full test set at inference, it is still worth comparing with it. We added RoPE in the experiments as requested: a figure with the new results is available at this link (https://figshare.com/s/f5938eb4836861483dd1) and it will be included in the updated version of our manuscript to be uploaded soon. We note that even though RoPE requires having access to the full test set at inference time, on top of fresh simulations $\lbrace x_i \rbrace_{1 \leq i \leq N_s}$ to compute an OT matrix, its performance is not better than our proposal.
> One possible explanation is that the posterior targeted by RoPE is not the true posterior under the well-specified model, since the coupling $\pi(x,y)$ returned by OT is not an approximation of the joint $p(x,y)$, therefore the mixture formulation from the RoPE paper cannot provide the true posterior distribution.
> NB: We have implemented the RoPE algorithm using code from a private communication with the authors of Wehenkel et al. (2025)
>
> ## New baselines and posterior level metrics
>
> Following your remarks, we added RoPE and two new baselines also present in (Wehenkel et al. 2025): NNPE and J-NPE, which do not leverage the calibration dataset; these results will be available in the Appendix of the new version of our manuscript.
>
> We have also included the standard C2ST metric for the Gaussian example.
>
> Thank you for bringing (Vetter, Julius, et al. "Effortless, Simulation-Efficient Bayesian Inference using Tabular Foundation Models) to our knowledge. We are currently going through the author’s codebase to see whether we can implement their method to work with our settings and tasks.
>
> ## Discussion about a recent multi-fidelity approach: Hikida et al. (2025)
>
> Hikida et al. (2025) use a MLMC principle, which requires the use of seed-matched simulations for two successive levels, which is a limitation discussed in their conclusion. In our setting, this means calibration data cannot be independent of the simulations. In particular, they cannot use ground truth data as calibration data, while our method can be used in all cases.
>
> ## Wall clock per-calibration size
>
> We understand your concerns about the overall computational cost of our method given the multiple components involved during training. We added new tables in the Appendix with the wall clock for each task and calibration set size. We reported the per-epoch cost as well as the total training time. See this link (https://figshare.com/s/0edfe75b333f8500a40b) for a preview.
>
> We confirm that our FMCPE is slower to train than the other baselines—while still remaining fully practical in our experiments. This additional cost is largely explained by the flow-sampling step (ODE integration), which currently dominates the training time. In our implementation, this component is not yet optimized: we rely on a straightforward explicit-Euler solver, whereas more advanced integrators (e.g., midpoint methods, higher-order adaptive solvers, or JAX-based JIT compilation) could substantially reduce the number of required steps and accelerate training.
> At inference time, however, FMCPE remains competitive with existing methods. In contrast, we observe that RoPE’s computational cost increases rapidly as the size of the calibration dataset grows, which is expected for an OT-based approach.

---

### Author Response · Authors · 2025-11-26
**General comment to all reviewers**

We thank the reviewers for their careful reading, interesting and overall positive comments. We are hopeful that they can easily be turned into improvements of our paper. While most reviewers mentioned the paper is well written and the main ideas interesting and well explained, we acknowledge the directions suggested to improve the writing and accessibility of the paper. Our response to specific comments from each reviewer is detailed afterwards.

To summarize, we addressed:
- Comments regarding our positioning as a misspecification or multifidelity approach. In short, we propose to emphasize the word “calibration” to clarify our presentation, for instance, we changed the title of our submission.
- We added comparisons of our proposal vs RoPE (Wehenkel et al. 2025), which as expected appears not to be fully adapted to the considered settings. Two other baselines NNPE and J-NPE were also considered.
- We added a detailed ablation study including two variants of our method to better illustrate the importance of each component of what we propose.
- We added training and inference computing times for all compared methods.

---

### Author Response · Authors · 2025-12-02
**Detailed summary for AC**

Dear AC, following the reviewers’ comments, we have updated our manuscript into a revised pdf, where the main changes, including new descriptions, 5 new/modified figures and 3 new tables, are highlighted in red. In addition, we provide below a summary of these changes and how we addressed the points raised by the reviewers.

1) **Misspecification vs multifidelity.** Although this appears as a minor terminology and semantic issue, 3 out of 4 reviewers commented on our positioning and suggested that we used the term multifidelity rather than misspecification. We provided a detailed response to explain that whatever the favored terminology it does not rule out the validity and relevance of our method and also why we think misspecification is the right term. In addition, we modified our presentation, to correct the potentially confusing wording that may have generated these comments. We also suggested to put more emphasis on the word “calibration”, e.g. in the title.

2) **Augmented benchmark.** Following the reviewers’ requests, we included **RoPE in our evaluation** and reported its performance in **Figure 2** for C2ST and Wasserstein distance, and in **Figure 6** for all metrics defined in Section 4.2.  KDE including RoPE are also displayed in **Figure 7**. For the Gaussian task, we now report posterior-level metrics in Figure 2 (instead of joint metrics used for the other tasks), as ground-truth posterior samples are available for this case.
One reviewer also noted that some baselines from (Wehenkel et al., 2025) were missing. We have therefore added Noisy NPE (**NNPE**) and Joint NPE (**J-NPE**) in **Figure 6** for the Pendulum and Wind/Light Tunnel tasks across all four metrics.
Finally, to improve readability, we now report the Gaussian-task results for C2ST, Wasserstein distance, and **MMD in Figure 5**.
3) **Ablation study.** One reviewer asked for an ablation study of our method. We reported in **Figure 4** the performance of our method and 3 variations :
- $T_{\Theta} + T_{X}$ - Sequential: The trainings of $T_{\Theta}$ and $T_X$ are done separately, we use $T_X$ to sample from the source distribution during the training of $T_{\Theta}$.
- $T_{\Theta}$ only: We only train $T_{\Theta}$ and use a source distribution of the form $ \pi(\mathbf{\theta} | \mathbf{y}) = \hat{p}_{\mathbf{\Theta}|\mathbf{X}}(\mathbf{\theta} | \mathbf{X} = \mathbf{y}) $, which is equivalent to setting $T_X = Id$ in our method.
- $T_{X}$ only: We only train $T_X$ on calibration data. The estimated posterior then becomes the NPE trained en simulation data evaluated on $\tilde{x} = T_X(y)$.

4) **Computational times.** Two reviewers also asked for training and inference time comparisons between our method and the baselines. We now report these results in **Table 1** and **Table 2**, which provide the training and inference times, respectively, for every combination of task and calibration size. **Table 3** further breaks down the training time of our method into three components: the gradient update, the ODE solve for $T_X$​, and NPE sampling.
While our method is more expensive to train, it remains practical, with training times under 6 minutes for all settings. Moreover, the dominant cost—the ODE-based flow sampling for $T_X$— can be substantially reduced with more efficient integrators or sampling strategies. At inference time, our method is faster than RoPE across all tasks.

---

### Meta-Review · Program_Chairs · 2026-01-25

**Summary:**

This paper tackles the problem of what is referred to as misspecification in simulation-based inference, where there is a mismatch between the characteristics of simulated and and real data. The paper assumes there is access to a small amount of real calibration samples and proposes a flow matching procedure to shift the posterior that was fit on simulated data.

**Reviewer Concerns:**

Reviewers had three main concerns:

First, multiple reviewers pointed out that they considered the problem setting to be one of multifidelity, not misspecification. The authors ultimately disagreed here, but conceded to changing the language to focus on calibration rather than misspecification.

Second (and partially stemming from the first concern), there was a persistent concern about missing baselines. To address this, the authors included results using RoPE from Wehenkel et al. 2025.

Finally, there were concerns about the cost of the method, particularly training-time costs. The authors conceded this point but pointed out that there were ways of reducing the cost and the proposed method was still cheaper in some ways than baselines.

**Reviewer Scores:**

Although the author response may have placated some of the reviewers' concerns, it is unlikely that all reviewers would have shifted their assessment from reject to accept. The suggested and promised changes to the paper constitute major revisions and changes, and consequently the paper should be edited to address the reviewers' concerns and resubmitted.

---

### Decision · Program_Chairs · 2026-01-26

Reject